# Hierarchical Gaussian Mixture based Task Generative Model for Robust Meta-Learning

**Yizhou Zhang**[1*], **Jingchao Ni**[2*‡], **Wei Cheng**[3], **Zhengzhang Chen**[3], **Liang Tong**[4*],
**Haifeng Chen**[3], **Yan Liu**[1]
[1]University of Southern California    [2]AWS AI Labs
[3]NEC Laboratories America    [4]Stellar Cyber Inc.
[1]{zhangyiz,yanliu.cs}@usc.edu;    [2]nijingchao@gmail.com;
[3]{weicheng,zchen,haifeng}@nec-labs.com;    [4]ltong@stellarcyber.ai

## Abstract

Meta-learning enables quick adaptation of machine learning models to new tasks with limited data. While tasks could come from varying distributions in reality, most of the existing meta-learning methods consider both training and testing tasks as from the same uni-component distribution, overlooking two critical needs of a practical solution: (1) the various sources of tasks may compose a multi-component mixture distribution, and (2) novel tasks may come from a distribution that is unseen during meta-training. In this paper, we demonstrate these two challenges can be solved jointly by modeling the density of task instances. We develop a meta-training framework underlain by a novel Hierarchical Gaussian Mixture based Task Generative Model (HTGM). HTGM extends the widely used empirical process of sampling tasks to a theoretical model, which learns task embeddings, fits the mixture distribution of tasks, and enables density-based scoring of novel tasks. The framework is agnostic to the encoder and scales well with large backbone networks. The model parameters are learned end-to-end by maximum likelihood estimation via an Expectation-Maximization (EM) algorithm. Extensive experiments on benchmark datasets indicate the effectiveness of our method for both sample classification and novel task detection.

## 1   Introduction

Training models in small data regimes is of fundamental importance. It demands a model's ability to quickly adapt to new environments and tasks. To compensate for the lack of training data for each task, meta-learning (*a.k.a.* learning to learn) has become an essential paradigm for model training by generalizing meta-knowledge across tasks [41, 8]. While most existing meta-learning approaches were built upon an assumption that all training/testing tasks are sampled from the same distribution, a more realistic scenario should accommodate training tasks that lie in a mixture of distributions, and testing tasks that may belong to or deviate from the learned distributions. For example, in recent medical research, a global model is typically trained on the historical medical records of a certain set of patients in the database [40, 49]. However, due to the uniqueness of individuals (*e.g.*, gender, age, genetics) [43], patients' data have a substantial discrepancy, and the pre-trained model may demonstrate significant demographic or geographical biases when testing on a new patient [35, 10]. This issue can be mitigated by personalized medicine approaches [5, 31], where each patient is regarded as a task, and the pre-trained model is fine-tuned (*i.e.*, personalized) on a support set of a few records collected in a short period (*e.g.*, a few weeks) from every patient for adaptation. In this case, the training tasks (*i.e.,* patients) could be sampled from a mixture of distributions (*e.g.*, different age

---

[*]This work was done primarily at NEC Laboratories America. [‡]Corresponding author.

groups), and a testing task may or may not belong to any of the observed groups. Similar examples can be found in other applications such as the detection of fake news dissemination [7, 55, 57], where a task is a post, whose support set consists of a few profiles of users who have disseminated it in a short period. The training posts may be drawn from a mixture of topics, and a testing post may not belong to any of the topics. As such, a meta-training strategy that is able to fit a model to a mixture of task distributions and enable the identification of novel tasks during inference time is desirable for making meta-learning a practical solution.

One way to tackle the mixture distribution of tasks is to tailor the transferable knowledge to each task by learning a task-specific representation [33, 48, 25], but as discussed in [52], the over-customized knowledge prevents its generalization among closely related tasks (*e.g.*, tasks from the same distribution). The more recent methods try to balance the generalization and customization of the meta-knowledge by promoting *local generalization* either among a cluster of related tasks [52], or within a neighborhood in a meta-knowledge graph of tasks [53]. Neither of them explicitly learns the underlying distribution from which the tasks are generated, rendering them infeasible for detecting novel tasks that are out-of-distribution. However, detecting novel tasks is crucial in high-stake domains, such as medicine and finance, which provides users (*e.g.*, physicians) confidence on whether to trust the results of a testing task or not, and facilitates downstream decision-making.

In [21], a task-specific tuning variable was introduced to modulate the initial parameters learned by the optimization-based meta-learning method MAML [8], so that the impacts of the meta-knowledge on different tasks are adjusted differently, *e.g.*, novel tasks are less influenced than the known tasks. Whereas, this method focuses on improving model performance on different tasks (either known or novel), but neglects the critical mission of detecting which tasks are novel. In practice, providing an unreliable accuracy on a novel task, without differentiating it from other tasks may be meaningless.

Since the aforementioned methods cannot simultaneously handle the mixture distribution of tasks and novel tasks, a practical solution is in demand. In this work, we consider tasks as instances, and demonstrate the dual problem of modeling the mixture of task distributions and detecting novel tasks are two sides of the same coin, *i.e.*, density estimation on task instances. To this end, we propose a $\underline{\text{H}}$ierarchical Gaussian Mixture based $\underline{\text{T}}$ask $\underline{\text{G}}$enerative $\underline{\text{M}}$odel (HTGM) to explicitly model the generative process of task instances. Our contributions are summarized as follows.

- We extended the widely used empirical process of generating tasks to a theoretical process specified by a hierarchy of Gaussian mixture (GM) distributions. HTGM generates a *task embedding* from a *task-level* GM, and uses it to define the task-conditioned mixture probabilities for a *class-level* GM, from which the data samples are drawn. To allow realistic classes per task, a new Gibbs distribution was proposed to underlie the class-level GM.

- HTGM is an encoder-agnostic framework, thus is flexible to different domains. It inherits metric-based meta-learning methods, and only introduces a small overhead to an encoder for parameterizing its distributions, thus is efficient for large backbone networks. The model parameters are learned end-to-end by maximum likelihood estimation via a principled EM algorithm. The bounds of our likelihood function were also theoretically analyzed.

- In the experiments, we evaluated HTGM on benchmark datasets regarding its scalability to large networks, effectiveness in modeling the mixture distribution of tasks, and usefulness in identifying novel tasks. The results demonstrate HTGM outperforms the state-of-the-art (SOTA) baselines with significant improvements in most cases.

## 2 Related Work

To the best of our knowledge, this is the first work to explicitly model the generative process of task instances from a mixture of distributions for meta-learning with novel task detection. Meta-learning aims to handle the few-shot learning problem, which derives memory-based [30], optimization-based [8, 26], and metric-based methods [47, 41], and often considers an artificial scenario where training/test tasks are sampled from the same distribution. To enable more varying tasks, task-adaptive methods facilitate the customization of meta-knowledge by learning task-specific parameters [37, 25], temperature scaling parameters [33], and task-specific modulation on model initialization [48, 52, 53, 21]. Among them, there are methods tackling the mixture distribution of tasks by clustering tasks [52, 13, 14, 17], learning task similarity and graphs [58, 53], and relocating the initial parameters for different tasks so that they use the meta-knowledge differently [21]. As discussed

before, none of these methods jointly handles the mixture of task distributions and the detection of novel tasks in meta-testing stage. A more detailed discussion is in Appendix B.1.

Our model is built upon metric-based methods and learns task embeddings to model task distributions. Achille et al. [1] proposed to learn embeddings for tasks and introduced a meta-learning method, but not for few-shot learning. Their embeddings are from a pre-specified set of tasks (rather than episode-wise sampling), and the meta-learning framework is for model selection. The model in [52] has an augmented encoder for task embedding, but it does not model task generation, and is not designed for novel task detection (an empirical comparison is in Sec. 4.1).

The conventional novelty detection aims to identify and reject samples from unseen classes [6]. It relates to open-set recognition [46], which aims to simultaneously identify unknown samples and classify samples from known classes. Out-of-distribution (OOD) detection [27, 28] can be seen as a special case of novelty detection where novel samples are from other problem domains or datasets, thus are considered to be easier to detect than novelties [6]. These methods are for large-scale training. In contrast, we want to detect novel tasks, which is a new problem in the small data regime.

Hierarchical Gaussian Mixture (HGM) model has appeared in some traditional works [9, 32, 3, 50] for hierarchical clustering by applying GM agglomeratively or divisively, which do not pre-train models for meta-learning, and is remarkably different from the topic in this paper. The differences are elaborated in Appendix B.2. Moreover, the idea of learning groups/clusters of tasks also appeared in some multi-task learning (MTL) models. The key difference between these methods and our method HTGM lies in the difference between MTL and meta-learning. In an MTL method, all tasks are known *a priori*, *i.e.*, the testing tasks are from the set of training tasks, and the model is inductive at the sample-level but non-inductive at the task-level. More discussions are in Appendix B.3.

## 3 Hierarchical Gaussian Mixture based Task Generative Model (HTGM)

### 3.1 Problem Statement

Meta-learning methods typically use an *episodic learning* strategy, where the meta-training set $\mathcal{D}^{\text{tr}}$ consists of a batch of episodes. Each episode samples a task $\tau$ from a distribution $p(\tau)$. Task $\tau$ has a support set $\mathcal{D}^{\text{s}}_\tau = \{(\mathbf{x}^{\text{s}}_i, y^{\text{s}}_i)\}^{n_{\text{s}}}_{i=1}$ for training, and a query set $\mathcal{D}^{\text{q}}_\tau = \{(\mathbf{x}^{\text{q}}_i, y^{\text{q}}_i)\}^{n_{\text{q}}}_{i=1}$ for testing, where $n_{\text{s}}$ is a small number to denote a few training samples. In particular, in a commonly used $N$-way $K$-shot $Q$-query task [47], $\mathcal{D}^{\text{s}}_\tau$ and $\mathcal{D}^{\text{q}}_\tau$ contain $N$ classes, with $K$ and $Q$ samples per class respectively, *i.e.*, $n_{\text{s}} = NK$ and $n_{\text{q}} = NQ$.

Let $f_{\boldsymbol{\theta}}(\mathbf{x}^*_i) \rightarrow y^*_i$ be a base model ($*$ denotes s or q), and $f_{\boldsymbol{\theta}}(\cdot; \mathcal{D}^{\text{s}}_\tau)$ be the adapted model on $\mathcal{D}^{\text{s}}_\tau$. The training objective on $\tau$ is to minimize the average test error of the adapted model, *i.e.*, $\mathbb{E}_{(\mathbf{x}^{\text{q}}_i, y^{\text{q}}_i) \in \mathcal{D}^{\text{q}}_\tau} \ell(y^{\text{q}}_i, f_{\boldsymbol{\theta}}(\mathbf{x}^{\text{q}}_i; \mathcal{D}^{\text{s}}_\tau))$, where $\ell(\cdot, \cdot)$ is a loss function (*e.g.*, cross-entropy loss), and the meta-training process aims to find the parameter $\boldsymbol{\theta}$ that minimizes this error over all episodes in $\mathcal{D}^{\text{tr}}$. Then, $f_{\boldsymbol{\theta}}$ is evaluated on every episode of a meta-test set $\mathcal{D}^{\text{te}}$ that samples a task from the same distribution $p(\tau)$. Usually, $p(\tau)$ is a simple distribution [8, 21]. In this work, $p(\tau)$ is generalized to a mixture distribution consisting of multiple components $p_1(\tau), ..., p_r(\tau)$, and a test episode may sample a task either in or out of any component of $p(\tau)$. As such, given the training tasks in $\mathcal{D}^{\text{tr}}$, our goal is to estimate the underlying density of $p(\tau)$, so that once a test task is given, we can (1) identify if it is a novel task, and (2) adapt $f_{\boldsymbol{\theta}}$ to it with optimal accuracy. Specifically, the base model $f_{\boldsymbol{\theta}}$ can be written as a combination of an encoder $g_{\boldsymbol{\theta}_e}$ and a predictor $h_{\boldsymbol{\theta}_p}$, *i.e.*, $f_{\boldsymbol{\theta}}(\mathbf{x}^*_i) = h_{\boldsymbol{\theta}_p}(g_{\boldsymbol{\theta}_e}(\mathbf{x}^*_i))$ [44]. In this work, we focus on a metric-based non-parametric learner, *i.e.*, $\boldsymbol{\theta}_p = \varnothing$ (*e.g.*, prototypical networks [41]), not only because metric-based classifiers were confirmed as more effective than probabilistic classifiers for novelty detection [12], but also for its better training efficiency that fits large backbone networks than the costly nested-loop training of optimization-based methods [44].

Formally, our goal is to find the model parameter $\boldsymbol{\theta}$ that maximizes the likelihood of observing a task $\tau$. In other words, let $f_{\boldsymbol{\theta}}(\mathbf{x}^*_i) = \mathbf{e}^*_i \in \mathbb{R}^d$ be sample embedding, we want to maximize the likelihood of the joint distribution $p_{\boldsymbol{\theta}}(\mathbf{e}^*_i, y^*_i)$ on the observed data in $\mathcal{D}_\tau = \{\mathcal{D}^{\text{s}}_\tau, \mathcal{D}^{\text{q}}_\tau\}$. We consider each task $\tau$ as an instance, with a representation $\mathbf{v}_\tau \in \mathbb{R}^d$ in the embedding space (the method to infer $\mathbf{v}_\tau$ is described in Sec. 3.3). To model the unobserved mixture component, we associate every task with a latent variable $z_\tau$ to indicate to which component it belongs. Suppose there are $r$ possible components, and let $n = n_{\text{s}} + n_{\text{q}}$ be the total number of samples in $\mathcal{D}_\tau$, the log-likelihood

to maximize can be written by hierarchically factorizing it on $y_i^*$ and marginalizing out $\mathbf{v}_\tau$ and $z_\tau$.

$$
\begin{aligned}
\ell(\mathcal{D}_\tau; \boldsymbol{\theta}) &= \frac{1}{n} \sum_{i=1}^{n} \log \left[ p_{\boldsymbol{\theta}}(\mathbf{e}_i^*, y_i^*) \right] = \frac{1}{n} \sum_{i=1}^{n} \log \left[ p_{\boldsymbol{\theta}}(\mathbf{e}_i^* | y_i^*) p(y_i^*) \right] \\
&= \frac{1}{n} \sum_{i=1}^{n} \log \left[ p_{\boldsymbol{\theta}}(\mathbf{e}_i^* | y_i^*) \Big[ \int_{\mathbf{v}_\tau} p(y_i^* | \mathbf{v}_\tau) p(\mathbf{v}_\tau) d\mathbf{v}_\tau \Big] \right] \\
&= \frac{1}{n} \sum_{i=1}^{n} \log \left[ p_{\boldsymbol{\theta}}(\mathbf{e}_i^* | y_i^*) \Big[ \int_{\mathbf{v}_\tau} p(y_i^* | \mathbf{v}_\tau) \Big[ \sum_{z_\tau=1}^{r} p(\mathbf{v}_\tau | z_\tau) p(z_\tau) \Big] d\mathbf{v}_\tau \Big] \right]
\end{aligned}
\tag{1}
$$

where $p_{\boldsymbol{\theta}}(\mathbf{e}_i^* | y_i^*)$ specifies the probability of sampling $\mathbf{e}_i^*$ from the $y_i^*$-th class, $p(y_i^* | \mathbf{v}_\tau)$ is the probability of sampling the $y_i^*$-th class for task $\tau$, and $p(\mathbf{v}_\tau | z_\tau)$ indicates the probability of generating a task $\tau$ from the $z_\tau$-th mixture component. $p(z_\tau)$ is a prior on the $z_\tau$-th component. Hence, Eq. (1) implies a generative process of task $\tau$: $z_\tau \to \mathbf{v}_\tau \to y_i^* \to \mathbf{e}_i^*$. Next, we define each of the aforementioned distributions and propose our HTGM method.

## 3.2  Model Specification and Parameterization

In Eq. (1), the *class-conditional distribution* $p_{\boldsymbol{\theta}}(\mathbf{e}_i^* | y_i^*)$, the *task-conditional distribution* $p(y_i^* | \mathbf{v}_\tau)$, and the *mixture distribution of tasks* defined by $\{ p(\mathbf{v}_\tau | z_\tau), p(z_\tau) \}$ are not specified. To make Eq. (1) optimizable, we introduce our HTGM that models the generative process of tasks. Because $\mathcal{D}_\tau^{\mathsf{s}}$ and $\mathcal{D}_\tau^{\mathsf{q}}$ follow the same distribution, in the following, we ignore the superscript $*$ for simplicity.

**Class-Conditional Distribution.** First, similar to [22, 23], we use Gaussian distribution to model the embeddings $\mathbf{e}_i$'s in every class. Let $\boldsymbol{\mu}_{y_i}^{\mathsf{c}}$ and $\boldsymbol{\Sigma}_{y_i}^{\mathsf{c}}$ be the mean and variance of the $y_i$-th class, then $p_{\boldsymbol{\theta}}(\mathbf{e}_i | y_i) = \mathcal{N}(\mathbf{e}_i | \boldsymbol{\mu}_{y_i}^{\mathsf{c}}, \boldsymbol{\Sigma}_{y_i}^{\mathsf{c}})$. In fact, the samples in all of the classes of task $\tau$ comprise a Gaussian mixture distribution, where $p(y_i)$ is the *mixture probability* of the $y_i$-th class. In Eq. (1), $p(y_i)$ is factorized to be task-specific, *i.e.*, $p(y_i | \mathbf{v}_\tau)$, which resorts to another mixture distribution $p(\mathbf{v}_\tau)$ of tasks, and establishes a structure of hierarchical mixture.

**Task-Conditional Distribution.** A straightforward definition of $p(y_i | \mathbf{v}_\tau)$ is the density at $\boldsymbol{\mu}_{y_i}^{\mathsf{c}}$ in a Gaussian distribution with $\mathbf{v}_\tau$ as the mean, where $\boldsymbol{\mu}_{y_i}^{\mathsf{c}}$ is the mean (or prototype) of the $y_i$-th class. However, doing so exposes two problems: (1) the density function of Gaussian distribution is log-concave with one global maximum. Given the mean and variance, maximizing its log-likelihood tends to collapse the prototypes $\boldsymbol{\mu}_{y_i}^{\mathsf{c}}$'s of all classes in $\tau$, making them indistinguishable and impairing classification; (2) given $\mathbf{v}_\tau$, this method tends to sample classes with small $D_{\mathbf{v}_\tau}(\boldsymbol{\mu}_{y_i}^{\mathsf{c}})$, where $D_{\mathbf{v}_\tau}(\cdot)$ measures the Mahalanobis distance between a data point and the Gaussian distribution centered at $\mathbf{v}_\tau$. However, in most of the existing works, classes are often uniformly sampled from a domain without any prior on distances [8]. Fitting the distance function with such "uniform" classes naively leads to an ill-posed learning problem with degenerated solutions. In light of these issues, we seek to define $p(y_i | \mathbf{v}_\tau)$ as a (parameterized) density function with at least $N$ global optimums so that it can distinguish the $N$ different class prototypes of $N$-way tasks. The $N$ equal (global) optimums also allow it to fit $N$ classes uniformly sampled from a domain. To this end, let $\boldsymbol{\mu}_k^{\mathsf{c}}$ be the *surrogate embedding* of the $k$-th class, we propose a Gibbs distribution $\pi(\boldsymbol{\mu}_k^{\mathsf{c}} | \mathbf{v}_\tau, \boldsymbol{\omega})$ defined by $\mathbf{v}_\tau$ and trainable parameters $\boldsymbol{\omega}$ with an energy function. Then we write $p(y_i = k | \mathbf{v}_\tau)$ as

$$
p_{\boldsymbol{\omega}}(y_i = k | \mathbf{v}_\tau) = \pi(\boldsymbol{\mu}_k^{\mathsf{c}} | \mathbf{v}_\tau, \boldsymbol{\omega}) = \frac{\exp\left[ -E_{\boldsymbol{\omega}}(\boldsymbol{\mu}_k^{\mathsf{c}}; \mathbf{v}_\tau) \right]}{\int_{\boldsymbol{\mu}_k^{\mathsf{c}}} \exp\left[ -E_{\boldsymbol{\omega}}(\boldsymbol{\mu}_k^{\mathsf{c}}; \mathbf{v}_\tau) \right]}
\tag{2}
$$

where $E_{\boldsymbol{\omega}}(\boldsymbol{\mu}_k^{\mathsf{c}}; \mathbf{v}_\tau) = \min\left( \{ \|\boldsymbol{\mu}_k^{\mathsf{c}} - \mathbf{W}_j \mathbf{v}_\tau\|_2^2 \}_{j=1}^N \right)$ is our energy function, and the denominator in Eq (2) is a normalizing constant (with respect to $\boldsymbol{\mu}_k^{\mathsf{c}}$), *a.k.a.* the partition function in an energy-based model (EBM) [20]. $\boldsymbol{\omega} = \{ \mathbf{W}_1, ..., \mathbf{W}_N \}$ are trainable parameters, with $\mathbf{W}_i \in \mathbb{R}^{d \times d}$. Given $\boldsymbol{\omega}$ and $\mathbf{v}_\tau$, Eq. (2) has $N$ global maximums at $\boldsymbol{\mu}_k^{\mathsf{c}} = \mathbf{W}_1 \mathbf{v}_\tau, ..., \boldsymbol{\mu}_k^{\mathsf{c}} = \mathbf{W}_N \mathbf{v}_\tau$. More interpretations of the proposed task-conditional distribution can be found in Appendix B.4.

**Mixture Distribution of Tasks.** In Eq. (1), the task distribution $p(\mathbf{v}_\tau)$ is factorized as a mixture of $p(\mathbf{v}_\tau | z_\tau = 1), ..., p(\mathbf{v}_\tau | z_\tau = r)$, weighted by their respective mixture probability $p(z_\tau)$. Thus we specify $p(\mathbf{v}_\tau)$ as a Gaussian mixture distribution. We introduce $\boldsymbol{\mu}_{z_\tau}^{\mathsf{t}}$ and $\boldsymbol{\Sigma}_{z_\tau}^{\mathsf{t}}$ as the mean and variance of each component, *i.e.*, $p(\mathbf{v}_\tau | z_\tau) = \mathcal{N}(\mathbf{v}_\tau | \boldsymbol{\mu}_{z_\tau}^{\mathsf{t}}, \boldsymbol{\Sigma}_{z_\tau}^{\mathsf{t}})$, and let $\boldsymbol{\rho} = [\rho_1, ..., \rho_r]$ be the mixture probabilities, where $\rho_r = p(z_\tau = r)$ and $\boldsymbol{\rho}$ can be Uniform$(r)$. Then $\mathbf{v}_\tau$ is genetrated in two

steps: (1) draw a latent task variable $z_\tau$ from a categorical distribution on $\boldsymbol{\rho}$, and (2) draw $\mathbf{v}_\tau$ from $\mathcal{N}(\boldsymbol{\mu}^t_{z_\tau}, \boldsymbol{\Sigma}^t_{z_\tau})$ [4]. As such, our HTGM generative process of an $N$-way $K$-shot $Q$-query task $\tau$ can be summarized as following, and Fig. 1 illustrates the corresponding graphical model:

1. Draw $z_\tau \sim \text{Categorical}([\rho_1, ..., \rho_r])$
2. Draw a task embedding $\mathbf{v}_\tau \sim \mathcal{N}(\boldsymbol{\mu}^t_{z_\tau}, \boldsymbol{\Sigma}^t_{z_\tau})$
3. For $k = 1, ..., N$:
   (a) Draw a class prototype $\boldsymbol{\mu}^c_k \sim \pi(\boldsymbol{\mu}^c_k | \mathbf{v}_\tau, \boldsymbol{\omega})$ from the proposed Gibbs distribution in Eq. (2)
   (b) For $i = 1, ..., K + Q$:
       i. Set $y_i = k$, draw a sample $\mathbf{e}_i \sim \mathcal{N}(\mathbf{e}_i | \boldsymbol{\mu}^c_{y_i}, \boldsymbol{\Sigma}^c_{y_i})$
       ii. Allocate $(\mathbf{e}_i, y_i)$ to the support set $\mathcal{D}^s_\tau$ if $i \leq K$; else allocate $(\mathbf{e}_i, y_i)$ to the query set $\mathcal{D}^q_\tau$

To reduce complexity, we investigate the feasibility of using isotropic Gaussian with tied variance, *i.e.*, $\boldsymbol{\Sigma}^c_1 = ... = \boldsymbol{\Sigma}^c_N = \sigma^2 \mathbf{I}$, for class distributions, which turned out to be efficient in our experiments. Here, $\mathbf{I}$ is an identity matrix, $\sigma$ is a hyperparameter. Tied variance is also a commonly used trick in Gaussian discriminate analysis (GDA) for generative classifiers [22, 39]. For task distributions, the variances $\boldsymbol{\Sigma}^t_1, ..., \boldsymbol{\Sigma}^t_r$ can be automatically inferred by our algorithm, as elaborated in Sec. 3.3.

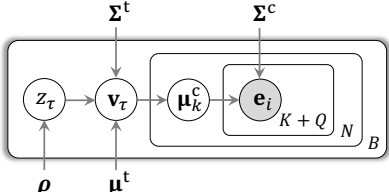

Finally, in Eq. (1), substituting $p_{\boldsymbol{\theta}}(\mathbf{e}_i | y_i) = \mathcal{N}(\mathbf{e}_i | \boldsymbol{\mu}^c_{y_i}, \sigma^2 \mathbf{I})$, $p_{\boldsymbol{\omega}}(y_i | \mathbf{v}_\tau) = \pi(\boldsymbol{\mu}^c_{y_i} | \mathbf{v}_\tau, \boldsymbol{\omega})$, $p(\mathbf{v}_\tau | z_\tau) = \mathcal{N}(\mathbf{v}_\tau | \boldsymbol{\mu}^t_{z_\tau}, \boldsymbol{\Sigma}^t_{z_\tau})$, whose probabilities are specified and parameterized, and letting $\boldsymbol{\rho} = \text{Uniform}(r)$ be a uniform prior, we get our

Figure 1: Graphical model of the proposed generative process for $B$ tasks with $N$ ways, $K$ shots and $Q$ queries.

HTGM induced loss $\ell_{\text{HTGM}}(\mathcal{D}_\tau; \boldsymbol{\theta}, \boldsymbol{\omega})$. The class means $\boldsymbol{\mu}^c_{y_i}$, task means $\boldsymbol{\mu}^t_{z_\tau}$ and variances $\boldsymbol{\Sigma}^t_{z_\tau}$ are inferred in the E-step of our EM algorithm (the details can be found in Sec. 3.3 and A.6).

### 3.3 Model Optimization

It is hard to directly optimize $\ell_{\text{HTGM}}(\mathcal{D}_\tau; \boldsymbol{\theta}, \boldsymbol{\omega})$, because the exact posterior inference is intractable (due to the integration over $\mathbf{v}_\tau$). To solve it, we resort to variational methods, and introduce an approximated posterior $q_{\boldsymbol{\phi}}(\mathbf{v}_\tau | \mathcal{D}^s_\tau)$, which is defined by an inference network $\boldsymbol{\phi}$, and implies we want to infer $\mathbf{v}_\tau$ from its observed support set $\mathcal{D}^s_\tau$. The query set $\mathcal{D}^q_\tau$ is not included because it is unavailable during model testing. Then we propose to maximize a lower-bound of Eq. (1), which is (the derivation can be found in Appendix A.1)

$$\ell_{\text{HTGM-L}}(\mathcal{D}_\tau; \boldsymbol{\theta}, \boldsymbol{\omega}) = \frac{1}{n} \sum_{i=1}^n \left( \log p_{\boldsymbol{\theta}, \boldsymbol{\omega}}(\mathbf{e}_i | y_i) + \mathbb{E}_{\mathbf{v}_\tau \sim q_{\boldsymbol{\phi}}(\mathbf{v}_\tau | \mathcal{D}^s_\tau)} [\log p_{\boldsymbol{\omega}}(y_i | \mathbf{v}_\tau) + \log \sum_{z_\tau = 1}^r p(\mathbf{v}_\tau | z_\tau) p(z_\tau)] \right)$$
$$+ H(q_{\boldsymbol{\phi}}(\mathbf{v}_\tau | \mathcal{D}^s_\tau))$$

(3)

where $H(q_{\boldsymbol{\phi}}(\mathbf{v}_\tau | \mathcal{D}^s_\tau)) = - \int_{\mathbf{v}_\tau} q_{\boldsymbol{\phi}}(\mathbf{v}_\tau | \mathcal{D}^s_\tau) \log q_{\boldsymbol{\phi}}(\mathbf{v}_\tau | \mathcal{D}^s_\tau) d\mathbf{v}_\tau$ is the entropy function. Similar to VAE [16], Eq. (3) estimates the expectation (in the second term) by sampling $\mathbf{v}_\tau$ from $q_{\boldsymbol{\phi}}(\mathbf{v}_\tau | \mathcal{D}^s_\tau)$, instead of the integration in Eq. (1), hence facilitates efficient computation. Next, we elaborate on the inference network, the challenges of maximizing Eq. (3), and our workarounds.

**Inference Network.** Similar to VAE, $q_{\boldsymbol{\phi}}(\mathbf{v}_\tau | \mathcal{D}^s_\tau)$ is defined as a Gaussian distribution $\mathcal{N}(\boldsymbol{\mu}^a_{z_\tau}, \bar{\sigma}^2 \mathbf{I})$, where $\boldsymbol{\mu}^a_{z_\tau}$ is the output of the inference network, which approximates $\boldsymbol{\mu}^t_{z_\tau}$ in Step 2 of the generative process, and $\bar{\sigma}$ is a hyperparameter for the corresponding variance. As illustrated by Fig. 2(a), the inference network is built upon the base model $f_{\boldsymbol{\theta}}(\cdot)$ with two non-parametric aggregation (*i.e.*, mean pooling) functions, thus $\boldsymbol{\phi} = \boldsymbol{\theta}$. The first function aggregates class-wise embeddings to prototypes $\boldsymbol{\mu}^c_{y_i}$'s, similar to prototypical networks [41]. Differently, the second aggregates all prototypes to $\boldsymbol{\mu}^a_{z_\tau}$. During model training, we used the reparameterization trick [16] to sample $\mathbf{v}_\tau$ from $\mathcal{N}(\boldsymbol{\mu}^a_{z_\tau}, \bar{\sigma}^2 \mathbf{I})$. It is noteworthy that $H(q_{\boldsymbol{\phi}}(\mathbf{v}_\tau | \mathcal{D}^s_\tau))$ in Eq. (3) becomes a constant because $\bar{\sigma}^2$ is a constant.

**Challenge 1: Trivial Solution.** In Eq. (3), since the first term $\log p_{\boldsymbol{\theta}, \boldsymbol{\omega}}(\mathbf{e}_i | y_i) = -\frac{1}{2\sigma^{2d}} \|\mathbf{e}_i - \boldsymbol{\mu}^c_{y_i}\|_2^2$ (constants were ignored) only penalizes the distance between a sample $\mathbf{e}_i$ and its own class mean $\boldsymbol{\mu}^c_{y_i}$ (*i.e.*, intra-class distances) without considering inter-class relationships, different class means $\boldsymbol{\mu}^c_1, ...,$

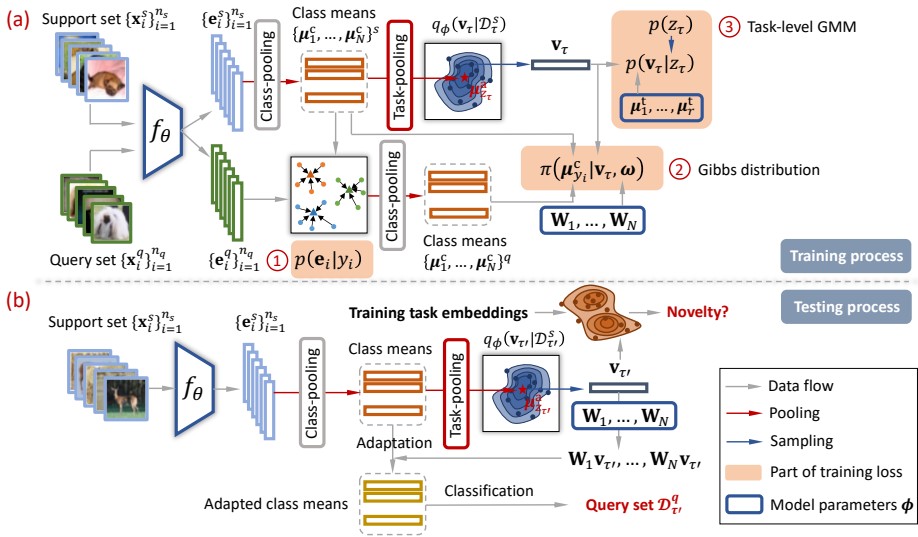

Figure 2: An illustration of HTGM on its (a) training process, and (b) testing process. In (a), ①②③ are the three parts of the training loss in Eq. (3). In (b), the training task embeddings contain the embeddings of all training tasks, *i.e.* the outputs of the task-pooling in (a).

$\boldsymbol{\mu}_N^c$ in task $\tau$ could collide, drawing all sample embeddings to the same spot. To avoid such a trivial solution and improve the stability of optimization, we apply negative sampling [29]

$$\ell_{\text{neg}}(\mathcal{D}_\tau; y_i, \boldsymbol{\theta}, \boldsymbol{\omega}) = -\log \mathbb{E}_{\mathbf{e}_j \sim \mathcal{D}_\tau} \Big[ \exp \big( -\frac{\|\mathbf{e}_j - \boldsymbol{\mu}_{y_i}^c\|_2^2}{2\sigma^{2d}} \big) \Big] \tag{4}$$

where $\mathbf{e}_j$ is a negative sample embedding from any class in the support set, and $\boldsymbol{\mu}_{y_i}^c$ is the mean of the positive class. In practice, we found it is beneficial to integrate $\ell_{\text{neg}}$ with our likelihood $\ell_{\text{HTGM}}$ in Eq. (1) during training, *i.e.* $\ell_{\text{HTGM}} + \frac{1}{n} \sum_{i=1}^n \ell_{\text{neg}}$. Correspondingly, from Eq. (3) we have

$$\ell(\mathcal{D}_\tau; \boldsymbol{\theta}, \boldsymbol{\omega}) = \ell_{\text{HTGM-L}}(\mathcal{D}_\tau; \boldsymbol{\theta}, \boldsymbol{\omega}) + \frac{1}{n} \sum_{i=1}^n \ell_{\text{neg}}(\mathcal{D}_\tau; y_i, \boldsymbol{\theta}, \boldsymbol{\omega}) \tag{5}$$

which does not only serve as a robust training loss, but also helps solve the next challenge.

**Challenge 2: The Partition Function in Eq. (2).** The second term $p_{\boldsymbol{\omega}}(y_i|\mathbf{v}_\tau)$ in Eq. (3) involves computing the partition function in Eq. (2) (*i.e.*, the denominator), which is intractable because of the integration over all possible $\boldsymbol{\mu}_k^c$'s. To solve it, we propose an upper bound of the partition function $\int_{\boldsymbol{\mu}_k} \exp \big[ -E_{\boldsymbol{\omega}}(\boldsymbol{\mu}_k^c; \mathbf{v}_\tau) \big] d\boldsymbol{\mu}_k^c \leq N\sqrt{\pi^d}$ (the derivation is in Appendix A.2), which is a constant with a specific $N$. By replacing the partition function in Eq. (2) with $N\sqrt{\pi^d}$, we got a lower bound of $p_{\boldsymbol{\omega}}(y_i|\mathbf{v}_\tau)$, which in turn relaxes the lower bound in Eq. (3). The following theorem (the proof is in Appendix A.3) states that the tightness of the relaxed bound is controllable.

**Theorem 3.1.** *Among the $N$ global maximums $\mathbf{W}_1\mathbf{v}_\tau, ..., \mathbf{W}_N\mathbf{v}_\tau$ of Eq. (2), let $\mathbf{W}_h\mathbf{v}_\tau$ and $\mathbf{W}_l\mathbf{v}_\tau$ $(1 \leq h, l \leq N)$ be the pair with the smallest Euclidean distance $D_{hl}$, we have*

$$\lim_{D_{hl} \to \infty} \int_{\boldsymbol{\mu}_k^c} \exp \big[ -E_{\boldsymbol{\omega}}(\boldsymbol{\mu}_k^c; \mathbf{v}_\tau) \big] d\boldsymbol{\mu}_k^c = N\sqrt{\pi^d} \tag{6}$$

This theorem indicates the partition function approximates $N\sqrt{\pi^d}$ when all pairs of the global maximums are far apart. It is noteworthy that when maximizing the likelihood, we fit $\mathbf{W}_1\mathbf{v}_\tau, ..., \mathbf{W}_N\mathbf{v}_\tau$ to different class prototypes $\boldsymbol{\mu}_1^c, ..., \boldsymbol{\mu}_N^c$ in $N$-way tasks. Because $\ell_{\text{neg}}$ in Eq. (4) tends to maximize the distances between different prototypes through the negative samples, maximizing the joint loss $\ell$ in Eq. (5) tends to separate $\mathbf{W}_1\mathbf{v}_\tau, ..., \mathbf{W}_N\mathbf{v}_\tau$, thus tighten the relaxed bound after using $N\sqrt{\pi^d}$ according to Theorem 3.1. This is another benefit of negative sampling.

**Optimization via Expectation-Maximization.** In the third term of $\ell_{\text{HTGM-L}}$ in Eq. (3), we need to estimate the mixture distribution $p(z_\tau)$. Similar to optimize Gaussian mixture models, we alternately

infer the posterior $p(z_\tau|\mathbf{v}_\tau)$ (*i.e.*, the mixture membership of $\mathbf{v}_\tau$) and solve the model parameters $\{\boldsymbol{\theta}, \boldsymbol{\omega}\}$ through an Expectation-Maximization algorithm. In E-step, we infer $p(z_\tau|\mathbf{v}_\tau)$ when fixing model parameters. In M-step, when fixing $p(z_\tau|\mathbf{v}_\tau)$, $\{\boldsymbol{\theta}, \boldsymbol{\omega}\}$ can be efficiently solved by optimizing Eq. (5) with stochastic gradient descent (SGD). The detailed optimization algorithm of HTGM can be found in Appendix A.6.

## 3.4 Model Adaptation

Fig. 2(b) illustrates the adaptation process of HTGM. Given a new $N$-way task $\tau'$ from the meta-test set $\mathcal{D}^{\text{te}}$, its support set $\mathcal{D}^{\text{s}}_{\tau'}$ is fed to the inference network to generate (1) class prototypes $\boldsymbol{\mu}^{\text{c}}_1$, ..., $\boldsymbol{\mu}^{\text{c}}_N$ [2] (similar to prototypical networks), and (2) distribution $q_\phi(\mathbf{v}_{\tau'}|\mathcal{D}^{\text{s}}_{\tau'})$, from which we draw the average task embedding $\mathbf{v}_{\tau'} = \boldsymbol{\mu}^{\text{a}}_{z_{\tau'}}$. Recall that the inference network is the base model $f_{\boldsymbol{\theta}}(\cdot)$ with class-pooling and task-pooling layers, as illustrated in Fig. 2(b), and $\phi = \boldsymbol{\theta}$. Then $\mathbf{v}_{\tau'}$ is projected to $\mathbf{W}_1\mathbf{v}_{\tau'}$, ..., $\mathbf{W}_N\mathbf{v}_{\tau'}$ which represent the $N$ optimal choices of class prototypes for task $\tau'$ as learned by the Gibbs distribution in Eq. (2) from the training tasks. They are used to adapt $\boldsymbol{\mu}^{\text{c}}_1$, ..., $\boldsymbol{\mu}^{\text{c}}_N$ so that the adapted prototypes are drawn towards the closest classes from the mixture component that task $\tau'$ belongs to. Specifically, the adaptation is performed by selecting the closest optimum for each prototype $\boldsymbol{\mu}^{\text{c}}_j$ ($1 \leq j \leq N$), that is

$$\bar{\boldsymbol{\mu}}^{\text{c}}_j = \alpha \boldsymbol{\mu}^{\text{c}}_j + (1-\alpha)\mathbf{W}_{l^*}\mathbf{v}_{\tau'}, \quad \text{where} \quad l^* = \underset{1 \leq l \leq N}{\arg\min} \, D(\boldsymbol{\mu}^{\text{c}}_j, \mathbf{W}_l\mathbf{v}_{\tau'}), \tag{7}$$

and $D(\cdot, \cdot)$ is the Euclidean distance, $\alpha$ is a hyperparameter. The following theorem confirms the effectiveness of the this adaptation method (the proof can be found in Appendix A.4).

**Theorem 3.2.** *The adapted prototypes $\bar{\boldsymbol{\mu}}^{\text{c}}_1$, ..., $\bar{\boldsymbol{\mu}}^{\text{c}}_N$ from Eq.* (7) *maximizes the lower-bound $\ell_{\text{HTGM-L}}$ in Eq.* (3) *of the likelihood in Eq.* (1) *when $\alpha = \frac{K}{K+2\sigma^{2d}}$.*

Theorem 3.2 suggests an automatic setup of $\alpha$, which can also be tuned empirically for optimal value using validation datasets. We evaluated the empirical values of $\alpha$ in our experiments and discussed their relationships with the theoretical values in Appendix A.4.

Finally, we (1) assess if $\tau'$ is a novelty by computing the likelihood of $\mathbf{v}_{\tau'}$ in a pre-fitted GMM on the embeddings $\mathbf{v}_\tau$'s of the training tasks in $\mathcal{D}^{\text{tr}}$, and (2) perform classification on each sample $\mathbf{x}'_i$ in the query set $\mathcal{D}^{\text{q}}_{\tau'}$ using the adapted prototypes by

$$p(y'_i = j'|\mathbf{x}'_i) = \frac{\exp\left(-D(f_{\boldsymbol{\theta}}(\mathbf{x}'_i), \bar{\boldsymbol{\mu}}^{\text{c}}_{j'})\right)}{\sum_{j=1}^{N} \exp\left(-D(f_{\boldsymbol{\theta}}(\mathbf{x}'_i), \bar{\boldsymbol{\mu}}^{\text{c}}_j)\right)} \tag{8}$$

which is the posterior probability. The derivation of Eq. (8) is in Appendix A.5.

## 4 Experiments

In this section, we evaluate HTGM's effectiveness on few-shot classification and novel task detection on benchmark datasets, and compare it with SOTA methods.

**Datasets.** The first dataset is the *Plain-Multi* benchmark [52]. It includes four fine-grained image classification datasets, *i.e.*, CUB-200-2011 (Bird), Describable Textures Dataset (Texture), FGVC of Aircraft (Aircraft), and FGVCx-Fungi (Fungi). In each episode, a task samples classes from one of the four datasets, so that different tasks are from a mixture of the four domains. The second dataset is the *Art-Multi* benchmark [53], whose distribution is more complex than Plain-Multi. Similar to [13], each image in Plain-Multi was applied with two filters, *i.e.*, *blur* filter and *pencil* filter, respectively, to simulate a changing distribution of few-shot tasks. Afterward, together with the original four datasets, a total of 12 datasets comprise the Art-Multi, and each task is sampled from one of them. Both benchmarks were divided into the meta-training, meta-validation, and meta-test sets by following their corresponding papers. Moreover, we used the Mini-ImageNet dataset [47] to evaluate the case of uni-component distribution of tasks, which is discussed in Appendix D.6.

**Baselines.** We compared HTGM with the relevant SOTA methods on meta-learning, including (1) optimization-based methods: MAML [8] and Meta-SGD [26] learn globally shared initialization

---

[2]In this section, $\boldsymbol{\mu}^{\text{c}}_j$ ($1 \leq j \leq N$) is the $j$-th class mean of support set, its superscript s is omitted for clarity.

| Setting | Model | Bird | Texture | Aircraft | Fungi | Avg. |
|---------|-------|------|---------|----------|-------|------|
| 5-way, 1-shot | TAML | 55.77±1.43 | 31.78±1.30 | 48.56±1.37 | 41.00±1.50 | 44.28 |
| | MAML | 53.94±1.45 | 31.66±1.31 | 51.37±1.38 | 42.12±1.36 | 44.77 |
| | Meta-SGD | 55.58±1.43 | 32.38±1.32 | 52.99±1.36 | 41.74±1.34 | 45.67 |
| | MUMOMAML | 56.82±1.49 | 33.81±1.36 | 53.14±1.39 | 42.22±1.40 | 46.50 |
| | HSML | 60.98±1.50 | 35.01±1.36 | 57.38±1.40 | 44.02±1.39 | 49.35 |
| | ARML | 62.33±1.47 | 35.65±1.40 | 58.56±1.41 | 44.82±1.38 | 50.34 |
| | ProtoNet | 61.54±1.27 | 38.84±1.42 | 73.42±1.23 | 46.52±1.42 | 55.08 |
| | MetaOptNet | 62.80±1.29 | 44.30±1.45 | 68.64±1.29 | 47.04±1.38 | 55.70 |
| | ProtoNet-Aug | 65.04±1.29 | 44.68±1.43 | 70.44±1.32 | 49.30±1.40 | 57.37 |
| | NCA | 62.58±1.25 | 40.98±1.44 | 68.70±1.26 | 46.36±1.34 | 54.66 |
| | FEATS | 62.60±1.31 | 44.12±1.49 | 68.86±1.28 | 47.92±1.34 | 55.88 |
| | HTGM (ours) | **70.12±1.28** | **47.76±1.49** | **75.52±1.24** | **52.06±1.41** | **61.37** |
| 5-way, 5-shot | TAML | 69.50±0.75 | 45.11±0.69 | 65.92±0.74 | 50.99±0.87 | 57.88 |
| | MAML | 68.52±0.79 | 44.56±0.68 | 66.18±0.71 | 51.85±0.85 | 57.78 |
| | Meta-SGD | 67.87±0.74 | 45.49±0.68 | 66.84±0.70 | 52.51±0.81 | 58.18 |
| | MUMOMAML | 70.49±0.76 | 45.89±0.69 | 67.31±0.68 | 53.96±0.82 | 59.41 |
| | HSML | 71.68±0.73 | 48.08±0.69 | 73.49±0.68 | 56.32±0.80 | 62.39 |
| | ARML | 73.34±0.70 | 49.67±0.67 | 74.88±0.64 | 57.55±0.82 | 63.86 |
| | ProtoNet | 78.88±0.72 | 57.93±0.75 | 86.42±0.57 | 62.52±0.79 | 71.44 |
| | MetaOptNet | 81.66±0.71 | **61.97±0.78** | 84.03±0.56 | 63.80±0.81 | 72.87 |
| | ProtoNet-Aug | 80.62±0.71 | 58.30±0.77 | 87.05±0.53 | 63.62±0.81 | 72.39 |
| | NCA | 79.16±0.75 | 58.69±0.76 | 85.27±0.53 | 61.68±0.80 | 71.20 |
| | FEATS | 78.37±0.72 | 57.02±0.73 | 85.55±0.54 | 61.56±0.80 | 70.63 |
| | HTGM (ours) | **82.27±0.74** | 60.67±0.78 | **88.48±0.52** | **65.70±0.79** | **74.28** |

Table 1: Results (accuracy±95% confidence) of the compared methods on Plain-Multi dataset.

among tasks. MUMOMAML [48] is a task-specific method. TAML [21] handles imbalanced tasks. HSML [52] and ARML [53] learn locally shared initial parameters in clusters of tasks and neighborhoods of a meta-graph of tasks, respectively; and (2) Metric-based methods: ProtoNet [41] is the prototypical network. MetaOptNet [24] uses an SVM classifier with kernel metrics. ProtoNet-Aug [42], FEATS [54] and NCA [19] were built upon ProtoNet by augmenting images (*e.g.*, rotation, jigsaw), adding prototype aggregator (*e.g.*, Transformer), and using contrastive training loss (instead of prototype-based loss), respectively. The detailed setup of these methods is in Appendix C.1.

**Implementation.** Following [44], the optimization-based baselines used the standard four-block convolutional layers as the base learner, and the metric-based methods used ResNet-12. The output dimension of these networks is 640 (MetaOptNet uses 16000 as in its paper). In our experiments, we observed the optimization-based methods reported out-of-memory errors when using ResNet-12, indicating their limitation in using large backbone networks. To test them on ResNet-12, we followed the ANIL method [36] by pre-training ResNet-12 via ProtoNet, freezing the encoder, and fine-tuning the last fully-connected layer. In this case, HSML and ARML cannot model the mixture task distribution properly as they require joint training of the encoder and other layers. The details are in Appendix D.5. For training, Adam optimizer was used. Each batch contains 4 tasks. Each model was trained with 20000 episodes. The learning rate of the metric-based methods was $1e^{-3}$. The learning rates for the inner- and outer-loops of the optimization-based methods were $1e^{-3}$ and $1e^{-4}$. The weight decay was $1e^{-4}$. For HTGM, we set $\sigma = 1.0$, $\bar{\sigma} = 0.1$, $\alpha = 0.5$ (0.9) for 1-shot (5-shot) tasks. The number of mixture components $r$ varies *w.r.t.* different datasets, and was grid-searched within $[2, 4, 8, 16, 32]$. All hyperparameters were set using the meta-validation sets.

### 4.1 Experimental Results

**Few-shot classification.** Following [44], we report the mean accuracy and 95% confidence interval of 1000 random tasks with 5-way 1-shot/5-shot, 5/25-query tests. Following [53], we report the accuracy of each domain and the overall average accuracy for Plain-Multi, and report the accuracy of each image filtering strategy and the overall average accuracy for Art-Multi. Table 1 and 2 summarize the results. From the tables, we have several observations. First, metric-based methods generally outperform optimization-based methods. This is because of the efficiency of metric-based methods, enabling them to fit a larger backbone network, which is consistent with the results in [44]. Built upon the metric-based method, HTGM only introduces a few distribution-related parameters and thus has the flexibility to scale with the encoder size. Second, the baselines designed for dealing

| Setting | Model | Original | Blur | Pencil | Avg. |
|---|---|---|---|---|---|
| | TAML | 42.22±1.39 | 40.02±1.41 | 35.11±1.34 | 39.11 |
| | MAML | 42.70±1.35 | 40.53±1.38 | 36.71±1.37 | 39.98 |
| | Meta-SGD | 44.21±1.38 | 42.36±1.39 | 37.21±1.39 | 41.26 |
| | MUMOMAML | 45.63±1.39 | 41.59±1.38 | 39.24±1.36 | 42.15 |
| 5-way, | HSML | 47.92±1.34 | 44.43±1.34 | 41.44±1.34 | 44.60 |
| 1-shot | ARML | 45.68±1.34 | 42.62±1.34 | 39.78±1.34 | 42.69 |
| | ProtoNet | 55.23±1.31 | 51.70±1.42 | 49.22±1.44 | 52.05 |
| | MetaOptNet | 56.10±1.35 | 52.33±1.43 | 49.08±1.45 | 52.50 |
| | ProtoNet-Aug | 57.63±1.34 | 55.00±1.40 | 49.73±1.53 | 54.12 |
| | NCA | 56.12±1.35 | 50.80±1.49 | 47.99±1.45 | 51.64 |
| | FEATS | 54.33±1.33 | 50.90±1.48 | 47.96±1.48 | 51.07 |
| | **HTGM (ours)** | **61.18±1.34** | **58.80±1.42** | **53.23±1.48** | **57.74** |
| | TAML | 58.54±0.73 | 55.23±0.75 | 49.23±0.75 | 54.33 |
| | MAML | 58.30±0.74 | 55.71±0.74 | 49.59±0.73 | 54.50 |
| | Meta-SGD | 57.82±0.72 | 55.54±0.73 | 50.24±0.72 | 54.53 |
| | MUMOMAML | 58.60±0.75 | 56.29±0.72 | 51.15±0.73 | 55.35 |
| 5-way, | HSML | 60.63±0.73 | 57.91±0.72 | 53.93±0.72 | 57.49 |
| 1-shot | ARML | 61.78±0.74 | 58.73±0.75 | 55.27±0.73 | 58.59 |
| | ProtoNet | 71.34±0.73 | 67.28±0.75 | 64.32±0.76 | 67.65 |
| | MetaOptNet | 72.33±0.72 | 68.90±0.78 | 63.89±0.71 | 68.37 |
| | ProtoNet-Aug | 72.87±0.71 | 70.50±0.72 | 63.98±0.73 | 68.78 |
| | NCA | 72.44±0.72 | 67.33±0.71 | 62.98±0.78 | 67.58 |
| | FEATS | 71.99±0.71 | 67.54±0.72 | 63.09±0.76 | 67.54 |
| | **HTGM (ours)** | **74.67±0.70** | **71.24±0.73** | **65.22±0.77** | **70.37** |

Table 2: Results (accuracy±95% confidence) of the compared methods on Art-Multi dataset.

with mixture distributions of tasks, *i.e.*, HSML and ARML, outperform their counterparts without such design, demonstrating the importance to consider mixture task distribution in practice. Finally, HTGM outperforms the SOTA baselines in most cases by large margins, suggesting its effectiveness in modeling the generative process of task instances.

**Novel task detection.** We also evaluate HTGM on the task of detecting novel $N$-way-$K$-shot tasks ($N = 5$, $K = 1$) that are drawn out of the training task distributions. To this end, we train each comapred model in the Original domain in Art-Multi dataset, and test the model on tasks drawn from either Original domain (*i.e.*, known tasks), or {Blur, Pencil} domains (*i.e.*, novel tasks), and evaluate if the model can tell whether a testing task is known or novel.

For comparison, since none of the baselines detects novel tasks, we adapt them as follows. For metric-based methods, since they use a fixed encoder for all training/testing tasks, we averaged the sample embeddings in each task to represent the task. Then a separate GMM model was built upon the training task embeddings, and its likelihood was adapted to score the novelty of testing tasks (some details of the setup are in Appendix C.2).

However, optimization-based models perform gradient descent on the support set of each task, leading to varying encoders per task. As such, sample embeddings of different tasks are not comparable, and we cannot obtain task

| Model | AUROC | AP | Max-F1 |
|---|---|---|---|
| HSML | 55.96 | 37.94 | 50.17 |
| ProtoNet | 65.17 | 41.51 | 56.07 |
| MetaOptNet | 72.71 | 63.77 | 58.33 |
| NCA | 66.28 | 51.45 | 52.74 |
| ProtoNet-Aug | 72.67 | 57.93 | 59.07 |
| FEATS | 59.35 | 42.57 | 49.31 |
| HTGM w/o GMM | 70.24 | 62.45 | 57.75 |
| HTGM-Gaussian | 74.06 | 66.18 | **60.62** |
| HTGM | **75.66** | **68.03** | 60.51 |

Table 3: Comparison between HTGM, its variants and applicable baselines on novel task detection.

embeddings in the same way as before. Among them, only HSML has an augmented task-level encoder for task embedding, allowing us to include it for comparison. For a fair comparison, our HTGM also trains a GMM on its task embeddings for detecting novel tasks. Moreover, two HTGM variants were included for ablation analysis to understand some design choices: (1) HTGM-Gaussian replaces the Gibbs distribution in Eq. (2) with a Gaussian distribution; (2) HTGM w/o GMM removes the task-level GM, *i.e.*, the third term in Eq. (3). The classification results of the ablation variants are in Appendix D.4. Following [6, 46, 56], we report Area Under ROC (AUROC), Average Precision (AP), and Max-F1. Table 3 summarizes the results, from which we observe HTGM outperforms all

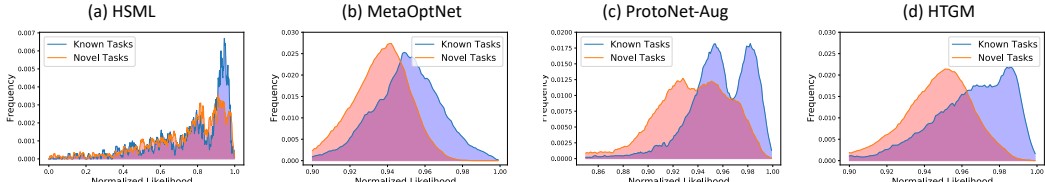

Figure 3: The frequency of tasks *w.r.t.* the normalized likelihood for (a) HSML (b) MetaOptNet (c) ProtoNet-Aug (d) HTGM. The x-axis ranges vary as only 95% tasks with top scores were preserved.

baselines over all evaluation metrics, indicating the superior quality of task embeddings learned by our model. The embeddings follow the specified mixture distribution of tasks $p(\mathbf{v}_\tau)$ as described in Sec. 3.2, which fits the mixture data well hence allowing to detect novel tasks that are close to the boundary. Since the baselines learn embeddings without explicit constraint, they don't fit the post-hoc GMM well. Moreover, HTGM outperforms HTGM w/o GMM, which is even worse than some other baselines. This further validates the necessity to introduce the regularization of task-level mixture distribution $p(\mathbf{v}_\tau)$. Also, the drops of AUROC and AP of HTGM-Gaussian demonstrate the importance of our unique design of the Gibbs distribution for the task-conditional distribution in Eq. (2). Similar to [46], in Fig. 3, we visualized the normalized likelihood histogram of known and novel tasks for HSML, MetaOptNet (the best baseline), ProtoNet-Aug (near-best baseline), and HTGM. To better interpret the figures, we calculated the ratio between the non-overlapped area of the two distributions and the total-area in Fig. 3 for HSML: 0.1379, MetaOpt: 0.4952, ProtoNet-Aug: 0.4806, and HTGM: 0.5578. As can be seen, the ratio of the non-overlapped area of HTGM is higher than other methods, which indicates the likelihoods (*i.e.*, novelty scores) of HTGM are more distinguishable for known and novel tasks than the baseline methods. We also analyzed the hyperparameters $\sigma$, $\bar{\sigma}$, and $r$ of HTGM in Appendix D.1, D.2, and D.3.

## 5   Conclusion

In this paper, we proposed a novel Hierarchical Gaussian Mixture based Task Generative Model (HTGM). HTGM models the generative process of task instances, and performs maximum likelihood estimation to learn task embeddings, which can help adjust prototypes acquired by the feature extractor and thus achieve better performance. Moreover, by explicitly modeling the distribution of tasks in the embedding space, HTGM can effectively detect the tasks that are drawn from distributions unseen during meta training. The extensive experimental results indicate the advantage of HTGM on both few-shot classification and novel task detection.

## 6   Broader Impact and Limitation

Our proposed method enables better novel task detection for meta-learning. In many areas requiring robust decisions, such as healthcare (as described in Sec. 1) and auto-driving, where the accuracy drop and uncertainty on novel tasks is inevitable, our model can raise an alarm to users (*e.g.*, doctors and human drivers) for diagnosis and decision-making. However, as a probabilistic machine learning model, HTGM does not guarantee 100% accuracy. Also, it is noteworthy that the entropy of $\pi(\boldsymbol{\mu}_{y_i}^c|\mathbf{v}_\tau, \boldsymbol{\omega})$ (in Eq. (2)) is proportional to the partition function (*i.e.* the denominator in Eq. (2)). Thus, our approximation in Sec. 3.3 that replaces the partition function with its upper bound increases the entropy, leading to increased noise in the inferred class prototypes. As such, in scenarios such as serious disease treatment and auto-driving in complex environments, to avoid potential wrong decisions from the model, human intervention is still necessary.

## Acknowledgments and Disclosure of Funding

This work was primarily finished during the internship of Yizhou Zhang (the first author) at NEC Laboratories America. Yizhou Zhang's work after the internship was partially supported by NSF Research Grant IIS-2226087 and the Annenberg Fellowship of the University of Southern California. We sincerely appreciate the comments and suggestions from the anonymous reviewers.

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

# A  Appendix for Details of Deriving HTGM

## A.1  The lower-bound of the likelihood function

In this section, we provide the details of the lower-bound in Eq. (3). By introducing the approximated posterior $q_\phi(\mathbf{v}_\tau | \mathcal{D}_\tau^\text{s})$, the likelihood in Eq. (1) becomes (the superscript $*$ is neglected for clarity)

$$
\begin{aligned}
\ell(\mathcal{D}_\tau, \boldsymbol{\theta}) &= \frac{1}{n} \sum_{i=1}^n \log p_{\boldsymbol{\theta}}(\mathbf{e}_i | y_i) + \frac{1}{n} \sum_{i=1}^n \log \Big( \int_{\mathbf{v}_\tau} p(y_i | \mathbf{v}_\tau) p(\mathbf{v}_\tau) d\mathbf{v}_\tau \Big) \\
&= \frac{1}{n} \sum_{i=1}^n \log p_{\boldsymbol{\theta}}(\mathbf{e}_i | y_i) + \frac{1}{n} \sum_{i=1}^n \log \Big( \int_{\mathbf{v}_\tau} p(y_i | \mathbf{v}_\tau) p(\mathbf{v}_\tau) \frac{q_\phi(\mathbf{v}_\tau | \mathcal{D}_\tau^\text{s})}{q_\phi(\mathbf{v}_\tau | \mathcal{D}_\tau^\text{s})} d\mathbf{v}_\tau \Big) \\
&= \frac{1}{n} \sum_{i=1}^n \log p_{\boldsymbol{\theta}}(\mathbf{e}_i | y_i) + \frac{1}{n} \sum_{i=1}^n \log \Big( \int_{\mathbf{v}_\tau} q_\phi(\mathbf{v}_\tau | \mathcal{D}_\tau^\text{s}) \frac{p(y_i | \mathbf{v}_\tau) p(\mathbf{v}_\tau)}{q_\phi(\mathbf{v}_\tau | \mathcal{D}_\tau^\text{s})} d\mathbf{v}_\tau \Big) \\
&\geq \frac{1}{n} \sum_{i=1}^n \log p_{\boldsymbol{\theta}}(\mathbf{e}_i | y_i) + \frac{1}{n} \sum_{i=1}^n \int_{\mathbf{v}_\tau} q_\phi(\mathbf{v}_\tau | \mathcal{D}_\tau^\text{s}) \Big[ \log p(y_i | \mathbf{v}_\tau) + \log p(\mathbf{v}_\tau) - \log q_\phi(\mathbf{v}_\tau | \mathcal{D}_\tau^\text{s}) \Big] d\mathbf{v}_\tau \\
&= \frac{1}{n} \sum_{i=1}^n \log p_{\boldsymbol{\theta}}(\mathbf{e}_i | y_i) + \frac{1}{n} \sum_{i=1}^n \int_{\mathbf{v}_\tau} q_\phi(\mathbf{v}_\tau | \mathcal{D}_\tau^\text{s}) \Big[ \log p(y_i | \mathbf{v}_\tau) + \log p(\mathbf{v}_\tau) \Big] d\mathbf{v}_\tau - \int_{\mathbf{v}_\tau} q_\phi(\mathbf{v}_\tau | \mathcal{D}_\tau^\text{s}) \log q_\phi(\mathbf{v}_\tau | \mathcal{D}_\tau^\text{s}) d\mathbf{v}_\tau \\
&= \frac{1}{n} \sum_{i=1}^n \log p_{\boldsymbol{\theta}}(\mathbf{e}_i | y_i) + \frac{1}{n} \sum_{i=1}^n \mathbb{E}_{\mathbf{v}_\tau \sim q_\phi(\mathbf{v}_\tau | \mathcal{D}_\tau^\text{s})} \Big[ \log p(y_i | \mathbf{v}_\tau) + \log p(\mathbf{v}_\tau) \Big] + H(q_\phi(\mathbf{v}_\tau | \mathcal{D}_\tau^\text{s})) \\
&= \frac{1}{n} \sum_{i=1}^n \log p_{\boldsymbol{\theta}}(\mathbf{e}_i | y_i) + \frac{1}{n} \sum_{i=1}^n \mathbb{E}_{\mathbf{v}_\tau \sim q_\phi(\mathbf{v}_\tau | \mathcal{D}_\tau^\text{s})} \Big[ \log p(y_i | \mathbf{v}_\tau) + \log \Big( \sum_{z_\tau = 1}^r p(\mathbf{v}_\tau | z_\tau) p(z_\tau) \Big) \Big] + H(q_\phi(\mathbf{v}_\tau | \mathcal{D}_\tau^\text{s}))
\end{aligned}
$$
(9)

where the fourth step uses Jensen's inequality. This completes the derivation of Eq. (3).

## A.2  The upper-bound of the partition function

In Sec. 3.3, we apply an upper bound on the partition function in Eq. (2) for solving the challenging 2. The derivation of the upper bound is as follows.

$$
\begin{aligned}
\int_{\boldsymbol{\mu}_{y_i}^\text{c}} \exp \big[ - E_{\boldsymbol{\omega}}(\boldsymbol{\mu}_{y_i}^\text{c}; \mathbf{v}_\tau) \big] d\boldsymbol{\mu}_{y_i}^\text{c} &= \int_{\boldsymbol{\mu}_{y_i}^\text{c}} \exp \big[ - \min \big( \{ ||\boldsymbol{\mu}_{y_i}^\text{c} - \mathbf{W}_j \mathbf{v}_\tau||_2^2 \}_{j=1}^N \big) \big] d\boldsymbol{\mu}_{y_i}^\text{c} \\
&= \int_{\boldsymbol{\mu}_{y_i}^\text{c}} \max \Big( \big\{ \exp \big[ - ||\boldsymbol{\mu}_{y_i}^\text{c} - \mathbf{W}_j \mathbf{v}_\tau||_2^2 \big] \big\}_{j=1}^N \Big) d\boldsymbol{\mu}_{y_i}^\text{c} < \int_{\boldsymbol{\mu}_{y_i}^\text{c}} \sum_{j=1}^N \exp \big[ - ||\boldsymbol{\mu}_{y_i}^\text{c} - \mathbf{W}_j \mathbf{v}_\tau||_2^2 \big] d\boldsymbol{\mu}_{y_i}^\text{c} \quad (10) \\
&= \sum_{j=1}^N \int_{\boldsymbol{\mu}_{y_i}^\text{c}} \exp \big[ - ||\boldsymbol{\mu}_{y_i}^\text{c} - \mathbf{W}_j \mathbf{v}_\tau||_2^2 \big] d\boldsymbol{\mu}_{y_i}^\text{c} = N \sqrt{\pi^d}
\end{aligned}
$$

where the last equation is from the multidimensional Gaussian integral. This completes the derivation of the upper bound of the partition function.

Note that replacing the partition function with its upper bound increases the noise in the inferred class mean. This is because that the entropy of distribution $\pi(\boldsymbol{\mu}_{y_i}^\text{c} | \mathbf{v}_\tau, \boldsymbol{\omega})$ ($\boldsymbol{\mu}_{y_i}^\text{c}$ is a class mean) in Eq. (2) is that:

$$
\begin{aligned}
H(\pi) &= - \int \pi(\boldsymbol{\mu}_{y_i}^\text{c} | \mathbf{v}_\tau, \boldsymbol{\omega}) \log \pi(\boldsymbol{\mu}_{y_i}^\text{c} | \mathbf{v}_\tau, \boldsymbol{\omega}) d\boldsymbol{\mu}_{y_i}^\text{c} \\
&= \int \pi(\boldsymbol{\mu}_{y_i}^\text{c} | \mathbf{v}_\tau, \boldsymbol{\omega}) \log Z d\boldsymbol{\mu}_{y_i}^\text{c} + \int \pi(\boldsymbol{\mu}_{y_i}^\text{c} | \mathbf{v}_\tau, \boldsymbol{\omega}) E_{\boldsymbol{\omega}}(\boldsymbol{\mu}_{y_i}^\text{c}; \mathbf{v}_\tau) \big] d\boldsymbol{\mu}_{y_i}^\text{c} \\
&= \log Z + \bar{E}_{\boldsymbol{\omega}}
\end{aligned}
$$
(11)

where $\bar{E}_{\boldsymbol{\omega}}$ is the average energy and $Z$ is the partition function. From this formula, we see that replacing $Z$ with its upper bound increases the entropy of $\pi(\boldsymbol{\mu}_{y_i}^\text{c} | \mathbf{v}_\tau, \boldsymbol{\omega})$, the distribution where $\boldsymbol{\mu}_{y_i}^\text{c}$ is sampled and/or inferred. Since entropy is a metric that measures the uncertainty of a random variable, increased entropy means that our estimation to $\mu_k^\text{c}$ is more uncertain, or in other words, more

noisy. This leads to a decreasing accuracy of meta-learning since the class mean $\boldsymbol{\mu}_{y_i}^{\mathtt{c}}$ is an important parameter to estimate in meta-learning. To alleviate this issue, we proposed to include an negative sampling term as in Eq. (4) to increase the distance between different class means so that the noise in the class mean estimation, which is brought by the upper bound approximation of $Z$, will not seriously influence the accuracy.

### A.3  The proof of Theorem 3.1

*Proof.* Let $B_j$ denote a sphere in $\mathbb{R}^d$. Its center is at $\mathbf{W}_j\mathbf{v}_\tau$ and its radius is $D_{hl}/2$. Because $\mathbf{W}_h\mathbf{v}_\tau$ and $\mathbf{W}_l\mathbf{v}_\tau$ $(1 \le h, l \le N)$ is the pair with the smallest Euclidean distance $D_{hl}$, for any pair of balls $B_j$ and $B_m$ we have $B_j \cap B_m$ is a null set (a set with 0 volume in $\mathbb{R}^d$).

In other words, there is no overlap between any pair of spheres. Therefore, if we compute the integral over the joint of all spheres, we have

$$\int_{\boldsymbol{\mu}_k^{\mathtt{c}} \in \cup_{m=1}^N B_m} \exp\big[-E_{\boldsymbol{\omega}}(\boldsymbol{\mu}_k^{\mathtt{c}}; \mathbf{v}_\tau)\big]d\boldsymbol{\mu}_k^{\mathtt{c}} = \sum_{m=1}^N \int_{\boldsymbol{\mu}_k^{\mathtt{c}} \in B_m} \exp\big[-E_{\boldsymbol{\omega}}(\boldsymbol{\mu}_k^{\mathtt{c}}; \mathbf{v}_\tau)\big]d\boldsymbol{\mu}_k^{\mathtt{c}} \tag{12}$$

Also, because there is no overlap between any pair of spheres, for each point $\boldsymbol{\mu}_k^{\mathtt{c}} \in B_m$, we have

$$-\min\left(\{\|\boldsymbol{\mu}_k^{\mathtt{c}} - \mathbf{W}_j\mathbf{v}_\tau\|_2^2\}_{j=1}^N\right) = -\|\boldsymbol{\mu}_k^{\mathtt{c}} - \mathbf{W}_m\mathbf{v}_\tau\|_2^2 \tag{13}$$

Therefore, we have the following derivation from Eq. (12).

$$\begin{aligned}
\int_{\boldsymbol{\mu}_k^{\mathtt{c}} \in \cup_{m=1}^N B_m} \exp\big[-E_{\boldsymbol{\omega}}(\boldsymbol{\mu}_k^{\mathtt{c}}; \mathbf{v}_\tau)\big]d\boldsymbol{\mu}_k^{\mathtt{c}} &= \sum_{m=1}^N \int_{\boldsymbol{\mu}_k^{\mathtt{c}} \in B_m} \exp\big[-E_{\boldsymbol{\omega}}(\boldsymbol{\mu}_k^{\mathtt{c}}; \mathbf{v}_\tau)\big]d\boldsymbol{\mu}_k^{\mathtt{c}} \\
&= \sum_{m=1}^N \int_{\boldsymbol{\mu}_k^{\mathtt{c}} \in B_m} \exp\big[-\|\boldsymbol{\mu}_k^{\mathtt{c}} - \mathbf{W}_m\mathbf{v}_\tau\|_2^2\big]d\boldsymbol{\mu}_k^{\mathtt{c}} = N\int_{\boldsymbol{\mu}_k \in B_m} \exp\big[-\|\boldsymbol{\mu}_k^{\mathtt{c}} - \mathbf{W}_m\mathbf{v}_\tau\|_2^2\big]d\boldsymbol{\mu}_k^{\mathtt{c}}
\end{aligned} \tag{14}$$

Meanwhile, since $\bigcup_{m=1}^N B_m$ is a sub-area of the entire $\mathbb{R}^d$ space, we have

$$\int_{\boldsymbol{\mu}_k^{\mathtt{c}} \in \cup_{m=1}^N B_m} \exp\big[-E_{\boldsymbol{\omega}}(\boldsymbol{\mu}_k^{\mathtt{c}}; \mathbf{v}_\tau)\big]d\boldsymbol{\mu}_k^{\mathtt{c}} \le \int_{\boldsymbol{\mu}_k^{\mathtt{c}}} \exp\big[-E_{\boldsymbol{\omega}}(\boldsymbol{\mu}_k^{\mathtt{c}}; \mathbf{v}_\tau)\big]d\boldsymbol{\mu}_k^{\mathtt{c}} \tag{15}$$

As $B_m$ is a sphere, we can convert the integral into spherical coordinates. Thus we have:

$$\begin{aligned}
\int_{\boldsymbol{\mu}_k \in B_m} \exp\big[-\|\boldsymbol{\mu}_k^{\mathtt{c}} - \mathbf{W}_m\mathbf{v}_\tau\|_2^2\big]d\boldsymbol{\mu}_k^{\mathtt{c}} &= \frac{2\sqrt{\pi^d}}{\Gamma(\frac{d}{2})}\int_0^{D_{hl}/2} \exp(-r^2)r^{d-1}dr \\
&= \frac{\sqrt{\pi^d}}{\Gamma(\frac{d}{2})}\int_0^{D_{hl}/2} \exp(-r^2)r^{d-2}(2rdr) \\
&= \frac{\sqrt{\pi^d}}{\Gamma(\frac{d}{2})}\int_0^{D_{hl}^2/4} \exp(-t)t^{\frac{d}{2}-1}dt \\
&= \frac{\sqrt{\pi^d}\gamma(\frac{d}{2}, D_{hl}^2/4)}{\Gamma(\frac{d}{2})}
\end{aligned} \tag{16}$$

where $\gamma(\cdot, \cdot)$ is lower incomplete gamma function and $\Gamma(\cdot)$ is gamma function ($\Gamma(x)$ the limitation of $\gamma(x, y)$ when $y \to +\infty$). According to the definition of lower incomplete gamma function, when $D_{hl}^2/4 \to +\infty$, we have

$$\lim_{D_{hl} \to \infty} \int_{\boldsymbol{\mu}_k^{\mathtt{c}} \in B_m} \exp\big[-E_{\boldsymbol{\omega}}(\boldsymbol{\mu}_k^{\mathtt{c}}; \mathbf{v}_\tau)\big]d\boldsymbol{\mu}_k^{\mathtt{c}} = \sqrt{\pi^d} \tag{17}$$

Therefore,

$$\lim_{D_{hl} \to \infty} \int_{\boldsymbol{\mu}_k^{\mathtt{c}}} \exp\big[-E_{\boldsymbol{\omega}}(\boldsymbol{\mu}_k^{\mathtt{c}}; \mathbf{v}_\tau)\big]d\boldsymbol{\mu}_k^{\mathtt{c}} \ge N\sqrt{\pi^d} \tag{18}$$

Since $N\sqrt{\pi^d}$ is its upper bound, based on the squeeze theorem, we have

$$\lim_{D_{hl}\to\infty}\int_{\boldsymbol{\mu}_k^c}\exp\big[-E_{\boldsymbol{\omega}}(\boldsymbol{\mu}_k^c;\mathbf{v}_\tau)\big]d\boldsymbol{\mu}_k^c=N\sqrt{\pi^d} \tag{19}$$

which completes the proof of Theorem 3.1. Moreover, from Equation 16, we know that the error ratio of the approximation, denoted as $AER$, can be bounded:

$$AER=\frac{N\sqrt{\pi^d}-\int_{\boldsymbol{\mu}_k^c}\exp\big[-E_{\boldsymbol{\omega}}(\boldsymbol{\mu}_k^c;\mathbf{v}_\tau)\big]d\boldsymbol{\mu}_k^c}{N\sqrt{\pi^d}}<1-\frac{\gamma(\frac{d}{2},D_{hl}^2/4)}{\Gamma(\frac{d}{2})}=1-\frac{\gamma(\frac{d}{2},D_{hl}^2/4)}{(\frac{d}{2}-1)!} \tag{20}$$

which monotonously decreases to 0 as $D_{hl}$ increases to $+\infty$ $\qquad\square$

### A.4 The proof of Theorem 3.2

Before starting the proof, we need to clarify the notations. When optimizing Eq. (3) in meta-training stage, we compute the class prototypes $\boldsymbol{\mu}_1^c,...,\boldsymbol{\mu}_N^c$ with both support sample means $\{\boldsymbol{\mu}_1^c,...,\boldsymbol{\mu}_N^c\}^s$ and query sample means $\{\boldsymbol{\mu}_1^c,...,\boldsymbol{\mu}_N^c\}^q$ (see Fig. 2). In Sec. 3.4, since during meta-testing stage we have no query samples, for clarity we omitted the notations to denote the $j$-th class mean of support set as $\boldsymbol{\mu}_j^c$. In the proof of the Theorem 3.2, we rigorously denote the support sample means of class $j$ as $\boldsymbol{\mu}_j^{c,s}$ and the query sample means of class $j$ as $\boldsymbol{\mu}_j^{c,q}$.

*Proof.* Recall the Eq. (3):

$$\ell_{\text{HTGM-L}}(\mathcal{D}_\tau;\boldsymbol{\theta},\boldsymbol{\omega})=\frac{1}{n}\sum_{i=1}^n\Big(\log p_{\boldsymbol{\theta},\boldsymbol{\omega}}(\mathbf{e}_i|y_i)+\mathbb{E}_{\mathbf{v}_\tau\sim q_\phi(\mathbf{v}_\tau|\mathcal{D}_\tau^s)}[\log p_{\boldsymbol{\omega}}(y_i|\mathbf{v}_\tau)+\log\sum_{z_\tau=1}^r p(\mathbf{v}_\tau|z_\tau)p(z_\tau)]\Big)$$
$$+H\big(q_\phi(\mathbf{v}_\tau|\mathcal{D}_\tau^s)\big) \tag{21}$$

By substituting the sampled $\mathbf{v}_\tau$ into it, we can acquire:

$$\ell_{\text{HTGM-L}}(\mathcal{D}_\tau;\boldsymbol{\theta},\boldsymbol{\omega})=\frac{1}{n}\sum_{i=1}^n\Big(\log p_{\boldsymbol{\theta},\boldsymbol{\omega}}(\mathbf{e}_i|y_i)+[\log p_{\boldsymbol{\omega}}(y_i|\mathbf{v}_\tau)+\log\sum_{z_\tau=1}^r p(\mathbf{v}_\tau|z_\tau)p(z_\tau)]\Big)$$
$$+H\big(q_\phi(\mathbf{v}_\tau|\mathcal{D}_\tau^s)\big) \tag{22}$$

Note that not every term in the Eq. (22) (the above likelihood function) contains class means $\boldsymbol{\mu}^c$. Only $\{p_{\boldsymbol{\omega}}(y_i|\mathbf{v}_\tau)\}$ and $\{p_{\boldsymbol{\theta},\boldsymbol{\omega}}(\mathbf{e}_i|y_i)\}$ are involved with $\boldsymbol{\mu}^c$. Thus, optimizing $\boldsymbol{\mu}^c$ to maximize $\ell_{\text{HTGM-L}}(\mathcal{D}_\tau;\boldsymbol{\theta},\boldsymbol{\omega})$ is equivalent to maximizing the following objective function $L(\boldsymbol{\mu}^c)$:

$$L(\boldsymbol{\mu}^c)=\sum_{j=1}^N\log p_{\boldsymbol{\omega}}(y_i=j|\mathbf{v}_\tau)+\sum_{i=1}^n\log p_{\boldsymbol{\theta},\boldsymbol{\omega}}(\mathbf{e}_i|y_i) \tag{23}$$

Moreover, since every term is involved with at most one individual $\boldsymbol{\mu}_j^c$, we can separately optimize each individual $\boldsymbol{\mu}_j^c$. We denote the embedding of the $i$-th sample in class $j$ as $\mathbf{e}_i^{(j)}$. The objective function for each individual $\boldsymbol{\mu}_j^c$ is:

$$L(\boldsymbol{\mu}_j^c)=\log p_{\boldsymbol{\omega}}(y_i=j|\mathbf{v}_\tau)+\sum_{i=1}^K\log p_{\boldsymbol{\theta},\boldsymbol{\omega}}(\mathbf{e}_i^{(j)}|y_i=j)$$
$$=\log\pi(\boldsymbol{\mu}_j^c|\mathbf{v}_\tau,\boldsymbol{\omega})+\sum_{i=1}^K\log\mathcal{N}(\mathbf{e}_i^{(j)}|\boldsymbol{\mu}_j^c,\sigma^2\mathbf{I}) \tag{24}$$
$$=-\frac{1}{2\sigma^{2d}}\sum_i^K||\mathbf{e}_i^{(j)}-\boldsymbol{\mu}_j^c||_2^2-\min_l||\mathbf{W}_l\mathbf{v}_\tau-\bar{\boldsymbol{\mu}}_j^c||_2^2+constant$$

The maximum likelihood estimation $\bar{\boldsymbol{\mu}}_j^c$ of $\boldsymbol{\mu}_j^c$ should maximize the above objective function. It is equivalent to find a solution set $\boldsymbol{\mu}_j^c=\bar{\boldsymbol{\mu}}_j^c, l=l^*$ that can minimize the following function:

$$L(\boldsymbol{\mu}_j^c,l)=\frac{1}{2\sigma^{2d}}\sum_i^K||\mathbf{e}_i^{(j)}-\boldsymbol{\mu}_j^c||_2^2+||\mathbf{W}_l\mathbf{v}_\tau-\boldsymbol{\mu}_j^c||_2^2 \tag{25}$$

We first compute the partial derivative of the above function with respect to $\boldsymbol{\mu}_j^{\mathsf{c}}$:

$$\frac{\partial L(\boldsymbol{\mu}_j^{\mathsf{c}}, l)}{\partial \boldsymbol{\mu}_j^{\mathsf{c}}} = \frac{1}{\sigma^{2d}} \sum_i^K (\boldsymbol{\mu}_j^{\mathsf{c}} - \mathbf{e}_i^{(j)}) + 2(\boldsymbol{\mu}_j^{\mathsf{c}} - \mathbf{W}_l \mathbf{v}_\tau) = \frac{1}{\sigma^{2d}} (K\boldsymbol{\mu}_j^{\mathsf{c}} - \sum_i^K \mathbf{e}_i^{(j)}) + 2(\boldsymbol{\mu}_j^{\mathsf{c}} - \mathbf{W}_l \mathbf{v}_\tau) \quad (26)$$

On the minimum, the partial derivative should be zero. Thus, $\bar{\boldsymbol{\mu}}_j^{\mathsf{c}} = \frac{1}{K+2\sigma^{2d}} \sum_i^K \mathbf{e}_i^{(j)} + \frac{2\sigma^{2d}}{K+2\sigma^{2d}} \mathbf{W}_l \mathbf{v}_\tau$ for the optimal $l^*$. Let us denote $\boldsymbol{\mu}_j^{\mathsf{c,s}} = \frac{1}{K} \sum_i^K \mathbf{e}_i^{(j)}$ (the support sample mean of class $j$), $\alpha = \frac{K}{K+2\sigma^{2d}}$ and $\mathbf{r}_l = \mathbf{W}_l \mathbf{v}_\tau - \boldsymbol{\mu}_j^{\mathsf{c}}$, then we have:

$$\bar{\boldsymbol{\mu}}_j^{\mathsf{c}} = \alpha \boldsymbol{\mu}_j^{\mathsf{c,s}} + (1-\alpha) \mathbf{W}_l \mathbf{v}_\tau = \boldsymbol{\mu}_j^{\mathsf{c,s}} + (1-\alpha)(\mathbf{W}_l \mathbf{v}_\tau - \boldsymbol{\mu}_j^{\mathsf{c,s}}) = \boldsymbol{\mu}_j^{\mathsf{c,s}} + (1-\alpha)\mathbf{r}_l \quad (27)$$

Therefore, we know that the optimal $\bar{\boldsymbol{\mu}}_j^{\mathsf{c}}, l$ satisfy the above equation. Because $l$ is an index between 1 and $N$, there are only $N$ solutions that satisfy the above equation. For these $N$ solutions, we have:

$$L(\bar{\boldsymbol{\mu}}_j^{\mathsf{c}}, l) = \frac{1}{2\sigma^{2d}} \sum_i^K ||\mathbf{e}_i^{(j)} - \bar{\boldsymbol{\mu}}_j^{\mathsf{c}}||_2^2 + ||\mathbf{W}_l \mathbf{v}_\tau - \bar{\boldsymbol{\mu}}_j^{\mathsf{c}}||_2^2$$

$$= \frac{1}{2\sigma^{2d}} \sum_i^K ||\mathbf{e}_i^{(j)} - \boldsymbol{\mu}_j^{\mathsf{c,s}} - \mathbf{r}_l||_2^2 + ||\mathbf{W}_l \mathbf{v}_\tau - \boldsymbol{\mu}_j^{\mathsf{c,s}} - (1-\alpha)\mathbf{r}_l||_2^2 \quad (28)$$

$$= \frac{1}{2\sigma^{2d}} (K||\mathbf{r}_l||_2^2 + \mathbf{r}_l \cdot \sum_i^K (\mathbf{e}_i^{(j)} - \boldsymbol{\mu}_j^{\mathsf{c,s}}) + \sum_i^K ||\mathbf{e}_i^{(j)} - \boldsymbol{\mu}_j^{\mathsf{c,s}}||_2^2) + \alpha^2 ||\mathbf{r}_l||_2^2$$

Note that $\boldsymbol{\mu}_j^{\mathsf{c,s}} = \frac{1}{K} \sum_i^K \mathbf{e}_i^{(j)}$. Thus $\sum_i^K (\mathbf{e}_i^{(j)} - \boldsymbol{\mu}_j^{\mathsf{c,s}}) = 0$. So we have:

$$L(\bar{\boldsymbol{\mu}}_j^{\mathsf{c}}, l) = \frac{K}{2\sigma^{2d}} ||\mathbf{r}_l||_2^2 + \alpha^2 ||\mathbf{r}_l||_2^2 + \text{const} \quad (29)$$

Therefore, we can know that the optimal $l$ that can minimize $L$ should minimize $||\mathbf{r}_l||_2^2$, which is $l^* = \arg\min_{1 \le l \le N} D(\boldsymbol{\mu}_j^{\mathsf{c}}, \mathbf{W}_l \mathbf{v}_{\tau'})$ using Euclidean distance $D(\cdot, \cdot)$. $\qquad\square$

However, our empirical evaluation shows that sometimes the emperically optimal value of $\alpha$ might be different from theoretically optimal value $\frac{K}{K+2\sigma^{2d}}$. For the 1-shot learning case, the optimal $\alpha$ we acquired from grid search is around 0.5 (same as theoretically optimal value). However, in the 5-shot learning case, the optimal $\alpha$ we acquired from grid search is around 0.9, slightly larger than 0.84 ($\frac{K}{K+2\sigma^{2d}}$). We suggests that it might be because during meta-training we replace the likelihood with its evidence lower bound and approximate the partition function with its upper bound, introducing noise into the generative model. Thus, we need to scale its weight down. Therefore, we propose to replace $\frac{K}{K+2\sigma^{2d}}$ with a hyperparameter $\alpha$ and fineunte it on validation set.

### A.5 Derivation of Eq. (8)

We first compute $p(y' = j'|\mathbf{x}')$ i.e. the posterior distribution of the query sample's label $y'$ conditioned on the sample. Then we select the label with the highest posterior probability. Utilizing the estimation of class means, we have the label conditional distribution:

$$p(y_i' = j'|\mathbf{x}_i') = \frac{p(y_i' = j', \mathbf{x}_i')}{\sum_{j=1}^N p(y_i' = j, \mathbf{x}_i')} = \frac{p(\mathbf{x}_i'|y_i' = j')p(y_i' = j')}{\sum_{j=1}^N p(\mathbf{x}_i'|y_i' = j)p(y_i' = j)} \quad (30)$$

Note that during the meta-testing stage, we should equally treat each class i.e., assume that the prior probability of each class should be same, i.e. $p(y_i' = 1) = ... = p(y_i' = N)$. Thus we have:

$$p(y_i' = j'|\mathbf{x}_i') = \frac{p(\mathbf{x}_i'|y_i' = j')}{\sum_{j=1}^N p(\mathbf{x}_i'|y_i' = j)} = \frac{\exp\left(-D(f_{\boldsymbol{\theta}}(\mathbf{x}_i'), \bar{\boldsymbol{\mu}}_{j'}^{\mathsf{c}})\right)}{\sum_{j=1}^N \exp\left(-D(f_{\boldsymbol{\theta}}(\mathbf{x}_i'), \bar{\boldsymbol{\mu}}_j^{\mathsf{c}})\right)} \quad (31)$$

### A.6 The training algorithm of HTGM

The training algorithm of HTGM is summarized in Algorithm 1.

**Algorithm 1:** Hierarchical Gaussian Mixture based Task Generative Model (HTGM)

---

**Input:** encoder $f_{\boldsymbol{\theta}}$, training dataset $\mathcal{D}^{\text{tr}}$, hyperparameters $r, \sigma, \bar{\sigma}$
**Output:** model parameters $\{\boldsymbol{\theta}, \boldsymbol{\omega}\}$

1   Pre-train the encoder $f_{\boldsymbol{\theta}}$ via ProtoNet with augmentations.

2   Pre-train the energy function in Eq. (2) by maximizing $\frac{1}{n} \sum_{i=1}^{n} \log p_{\boldsymbol{\theta}, \boldsymbol{\omega}}(\mathbf{e}_i | y_i) + \log p_{\boldsymbol{\omega}}(y_i | \mathbf{v}_\tau)$

3   **for** $i \leftarrow 1$ *to MaxEpoch* **do**

     /* E-step                                                              */

4      $\mathcal{V} = \varnothing$

5      **for** $\{\mathcal{D}_\tau^s = \{(\mathbf{x}_i^s, y_i^s)\}_{i=1}^{ns}, \mathcal{D}_\tau^q = \{(\mathbf{x}_i^q, y_i^q)\}_{i=1}^{nq}\}$ *in Dataloader*$(\mathcal{D}^{tr})$ **do**

         /* load a task episode                                         */

6          $\{\mathbf{e}_i^s\}_{i=1}^{ns} = \{f_{\boldsymbol{\theta}}(\mathbf{x}_i^s)\}_{i=1}^{ns}$ ;                     // embeddings of the support set

7          $\boldsymbol{\mu}_{z_\tau}^{\text{a}} = \text{Task-Pooling}(\text{Class-Pooling}(\{(\mathbf{e}_i^s, y_i^s)\}_{i=1}^{ns}))$ ;       // the mean of $q_\phi(\mathbf{v}_\tau | \mathcal{D}_\tau^s)$

8          Sample a task embedding $\mathbf{v}_\tau$ from $q_\phi(\mathbf{v}_\tau | \mathcal{D}_\tau^s) = \mathcal{N}(\boldsymbol{\mu}_{z_\tau}^{\text{a}}, \bar{\sigma}^2 \mathbf{I})$

9          $\mathcal{V} = \mathcal{V} \cup \{\mathbf{v}_\tau\}$

10      **end**

11      $\{z_\tau\}_{\tau=1}^{|\mathcal{V}|}, \{\boldsymbol{\mu}_1^{\text{t}}, ..., \boldsymbol{\mu}_r^{\text{t}}, \boldsymbol{\Sigma}_1^{\text{t}}, ..., \boldsymbol{\Sigma}_r^{\text{t}}\} = \text{GMM}(\mathcal{V})$. ;       // fit a GMM to $\mathcal{V}$, where $\{z_\tau\}_{\tau=1}^{|\mathcal{V}|}$ represents the labeling of the $\mathbf{v}_\tau$'s in $\mathcal{V}$

     /* M-step                                                              */

12      **for** $\{\mathcal{D}_\tau^s = \{(\mathbf{x}_i^s, y_i^s)\}_{i=1}^{ns}, \mathcal{D}_\tau^q = \{(\mathbf{x}_i^q, y_i^q)\}_{i=1}^{nq}\}$ *in Dataloader*$(\mathcal{D}^{tr})$ **do**

         /* load a task episode                                         */

13          $\{\mathbf{e}_i^s\}_{i=1}^{ns} = \{f_{\boldsymbol{\theta}}(\mathbf{x}_i^s)\}_{i=1}^{ns}$ ;                        // forward pass

14          $\{\mathbf{e}_i^q\}_{i=1}^{nq} = \{f_{\boldsymbol{\theta}}(\mathbf{x}_i^q)\}_{i=1}^{nq}$ ;                        // forward pass

15          $\{\boldsymbol{\mu}_1^{\text{c}}, ..., \boldsymbol{\mu}_N^{\text{c}}\}^{\text{s}} = \text{Class-Pooling}(\{(\mathbf{e}_i^s, y_i^s)\}_{i=1}^{ns})$

16          $\boldsymbol{\mu}_{z_\tau}^{\text{a}} = \text{Task-Pooling}(\{\boldsymbol{\mu}_1^{\text{c}}, ..., \boldsymbol{\mu}_N^{\text{c}}\}^{\text{s}})$ ;       // the mean of $q_\phi(\mathbf{v}_\tau | \mathcal{D}_\tau^s)$

17          Sample a task embedding $\mathbf{v}_\tau$ from $q_\phi(\mathbf{v}_\tau | \mathcal{D}_\tau^s) = \mathcal{N}(\boldsymbol{\mu}_{z_\tau}^{\text{a}}, \bar{\sigma}^2 \mathbf{I})$

18          **for** $j = 1, ..., N$ **do**

19             $\bar{\boldsymbol{\mu}}_j^{\text{c}} = \alpha \boldsymbol{\mu}_j^{\text{c}} + (1 - \alpha) \mathbf{W}_{l^*} \mathbf{v}_{\tau'}$ where $l^* = \arg\min_{1 \leq l \leq N} D(\boldsymbol{\mu}_j^{\text{c}}, \mathbf{W}_l \mathbf{v}_{\tau'})$

20          **end**

21          Calculate $\ell(\{\mathbf{e}_i^q\}_{i=1}^{nq}, \mathcal{V}, \{\bar{\boldsymbol{\mu}}_j^{\text{c}}\}_{j=1}^{N}, \{\boldsymbol{\mu}_1^{\text{t}}, ..., \boldsymbol{\mu}_r^{\text{t}}, \boldsymbol{\Sigma}_1^{\text{t}}, ..., \boldsymbol{\Sigma}_r^{\text{t}}\}, \sigma, \boldsymbol{\omega})$ ;   // calculate the loss in Eq. (5) using Eq. (3) and Eq. (4)

22          $\boldsymbol{\theta}, \boldsymbol{\omega} = \text{SGD}(\ell, \boldsymbol{\theta}, \boldsymbol{\omega})$ ;                    // update model parameters

23      **end**

24 **end**

---

# B   Appendix for Further Discussion

## B.1   Discussion about the novel task discussion and meta-learning

As we discussed in Sec. 2, to the best of our knowledge, our proposed method HTGM is the first work that jointly considers the task mixture distribution and novel task detection in meta-testing stage. There are some works considering how to identify novel task clusters in meta-training stage based on task embedding [52] or task likelihood [13]. However, they have their own respective drawbacks when handling novel task detection in meta-testing stage. For task-embedding-based method like [52], it does not explicitly model the task distribution. Instead, it considers how to model the task membership of the learnt clusters. As a result, they can only identify the outlying task clusters rather than individual novel tasks. However, in meta-testing stage, we expect the model to identify each individual novel task and raise alerts. The task-likelihood-based method DPMM [13] can handle individual novel tasks. However, it is hard for them to simultaneously handle quick detection and adaptation. This is because its likelihood was built on the entire model parameters, leading to model-dependent and time consuming computation. It is not a big issue for meta-training, but will serious limit its application to streaming tasks in meta-testing (e.g., in auto-driving domain) where efficiency is critical for timely alarms of novel tasks.

## B.2 Discussion about the relationship between HTGM and HGM model

To the best of our knowledge, the Hierarchical Gaussian Mixture (HGM) model has appeared in the traditional works [9, 32, 3, 50] for hierarchical clustering by applying Gaussian Mixture model agglomeratively or divisively on the input samples. They are unsupervised methods that infer clusters of samples, but do not pre-train embedding models (or parameter initializations) that could be fine-tuned for the adaptation to new tasks in meta-learning. Therefore, these methods are remarkably different from meta-learning methods, and we think it is a non-trivial problem to adapt the concept of HGM to solve the meta-learning problem. To this end, we need to (1) identify the motivation; and (2) solve the new technical challenges. For (1), the hierarchical structure of mixture distributions naturally appears when we want to model the generative process of tasks from a mixture of distributions, where each task contains another mixture distribution of classes (as suggested by Eq. (1)). In other words, the motivating point of our method is more on meta-learning than HGM. However, drawing such a connection between meta-learning and HGM is a novel contribution. For (2), our method is different from traditional HGM in (a) its generative process of tasks (Sec. 3.2), which is a theoretical extension of the widely used empirical process of generating tasks in meta-learning; (b) its Gibbs-style task-conditional distribution (Eq. (2)) for fitting uniformly sampled classes; (c) the metric-based end-to-end meta-learning framework (Fig. 2) (note the traditional HGM is not for learning embeddings); (d) the non-trivial derivation of the optimization algorithm in Sect. 3.3 and Alg. 1; and (e) the novel model adaptation process in Sec. 3.4. Solving the technical challenges in the new generative model is another novel contribution of the proposed method.

## B.3 Discussion about the related multi-task learning methods

The modeling of the clustering/grouping structure of tasks or the mixture of distributions of tasks has been studied in multi-tasking learning (MTL). In [51, 11], tasks are assumed to have a clustering structure, and the model parameters of the tasks in the same cluster are drawn to each other via optimization on their L2 distances. In [15], a subspace based regularization framework was proposed for grouping task-specific model parameters, where the tasks in the same group are assumed to lie in the same low dimensional subspace for parameter sharing. The method in [18] also uses the subspace based sharing of task parameters, but allows two tasks from different groups to overlap by having one or more bases in common. The method in [34] introduces a generative model for task-specific model parameters that encourages parameter sharing by modeling the latent mixture distribution of the parameters via the Dirichlet process and Beta process.

The key difference between these methods and our method HTGM lies in the difference between MTL and meta-learning. In an MTL method, all tasks are known *a priori*, *i.e.*, the testing tasks are from the set of training tasks, and the model is non-inductive at the task-level (but it is inductive at the sample-level). In HTGM, testing tasks can be disjoint from the set of training tasks, thus the model is inductive at the task-level. In particular, we aim to allow testing tasks that are not from the distribution of the training tasks by enabling the detection of novel tasks, which is an extension of the task-level inductive model. The second difference lies in the generative process. The method in [34] models the generative process of the task-specific model parameters (*e.g.*, the weights in a regressor). In contrast, HTGM models the generative process of each task by generating the classes in it, and the samples in the classes hierarchically, *i.e.*, the $(\mathbf{x}, y)$'s (in Eq. (1) and Sec. 3.2). In this process, we allow our model to fit uniformly sampled classes given a task (without specifying a prior on the distance function on classes) by the proposed Gibbs distribution in Eq. (2). Other remarkable differences to the aforementioned MTL methods include the inference network (Fig. 2(b)), which allows the inductive inference on task embeddings and class prototypes; the optimization algorithm (Sec. 3.3) to our specific loss function in Eq. (3), which is from the likelihood in Eq. (1); and the model adaptation algorithm (Sec. 3.4) for performing predictions in a testing task, and detecting novel tasks. As such, the MTL methods can not be trivially applied to solve our problem.

## B.4 Further interpretation of the task-conditional distribution

The task-conditional class distribution $p_{\boldsymbol{\omega}}(y_i = k|\mathbf{v}_\tau)$ in Eq. (2) is defined through an energy function $E_{\boldsymbol{\omega}}(\boldsymbol{\mu}_k^{\mathrm{c}}; \mathbf{v}_\tau) = \min (\{||\boldsymbol{\mu}_k^{\mathrm{c}} - \mathbf{W}_j \mathbf{v}_\tau||_2^2\}_{j=1}^N)$ with trainable parameters $\boldsymbol{\omega} = \{\mathbf{W}_1, ..., \mathbf{W}_N\}$, for allowing uniformly sampled classes per task. The conditional distribution $p(y_i|\mathbf{v}_\tau)$ represents how classes distribute for a given task $\tau$. The reason for its definition in Eq. (2) is as follows. If it is a Gaussian distribution with $\mathbf{v}_\tau$ (*i.e.*, task embedding) as the mean, $p(y_i = k|\mathbf{v}_\tau)$ can be interpreted as

the density at the representation of the $k$-th class in this Gaussian distribution, *i.e.*, the density at $\boldsymbol{\mu}_k$, which is the mean/surrogate embedding of the $k$-th class. One problem of this Gaussian $p(y_i|\mathbf{v}_\tau)$ is that different classes, *i.e.*, different $\boldsymbol{\mu}_{y_i}$'s, are not uniformly distributed, contradicting the practice that given a dataset (e.g., images), classes are often uniformly sampled for constituting a task in the empirical studies. Using a uniformly sampled set of classes to fit the Gaussian distribution $p(y_i|\mathbf{v}_\tau)$ will lead to an ill-posed learning problem, as described in Sec. 3.2. To solve it, we introduced $\boldsymbol{\omega} = \{\mathbf{W}_1, ..., \mathbf{W}_N\}$ in the energy function $E_{\boldsymbol{\omega}}(\boldsymbol{\mu}_k^c; \mathbf{v}_\tau)$ in Eq. (2). $\mathbf{W}_j \in \mathbb{R}^{d \times d}$ $(1 \le j \le N)$ can be interpreted as projecting $\mathbf{v}_\tau$ to the $j$-th space spanned by the basis (*i.e.*, columns) of $\mathbf{W}_j$. There are $N$ different spaces for $j = 1, ..., N$. Thus, the $N$ projected task means $\mathbf{W}_1\mathbf{v}_\tau, ..., \mathbf{W}_N\mathbf{v}_\tau$ are in $N$ different spaces. Fitting the energy function $E_{\boldsymbol{\omega}}(\boldsymbol{\mu}_k^c; \mathbf{v}_\tau)$ to $N$ uniformly sampled classes $\boldsymbol{\mu}_1^c, ..., \boldsymbol{\mu}_N^c$, which tend to be far from each other because they are uniformly random, tends to learn $\mathbf{W}_1, ..., \mathbf{W}_N$ that project $\mathbf{v}_\tau$ to $N$ far apart spaces that fit each of the $\boldsymbol{\mu}_1^c, ..., \boldsymbol{\mu}_N^c$ by closeness, due to the min-pooling operation. This mitigates the aforementioned ill-posed learning problem.

## C  Appendix for Implementation Details

### C.1  The setup of the compared models

**Encoder of Metric-based Meta-Learning.** For fairness, for all metric-based methods, including ProtoNet [41], MetaOptNet [24], ProtoNet-Aug [42], FEATS [54] and NCA [19], following [44, 24], we apply ResNet-12 as the encoder. ResNet-12 has 4 residual blocks, each has 3 convolutional layers with a kernel size of $3 \times 3$. ResNet-12 uses dropblock as a regularizer, and its number of filters is (60, 160, 320, 640). For MetaOptNet, following its paper [24], we flattened the output of the last convolutional layer to acquire a 16000-dimensional feature as the image embedding. For other baselines, following [44], we used a global average-pooling layer on the top of the last residual block to acquire a 640-dimensional feature as the image embedding.

**Further Details.** Following [41], ProtoNet, ProtoNet-Aug, and NCA use Adam optimizer with $\beta_1 = 0.9$ and $\beta_2 = 0.99$. We did grid-search for the initial learning rate of the Adam within $\{1e^{-2}, 1e^{-3}, 1e^{-4}\}$, where $1e^{-3}$ was selected, which is the same as the official implementation provided by the authors. For FEATS, we chose transformer as the set-to-set function based on the results reported by [54]. When pre-training the encoder in FEATS, following its paper [54], we applied the same setting as ProtoNet, which is to use Adam optimizer with an initial learning rate of $1e^{-3}$, $\beta_1 = 0.9$ and $\beta_2 = 0.99$. When training its aggregation function, we grid-searched the initial learning rate in $\{1e^{-4}, 5e^{-4}, 1e^{-5}\}$ since a larger learning rate leads to invalid results on our datasets. The optimal choice is $1e^{-4}$. For MetaOptNet, following its paper [24], we used SGD with Nesterov momentum of 0.9, an initial learning rate of 0.1 and a scheduler to optimize it, and applied the quadratic programming solver OptNet [2] for the SVM solution in it.

### C.2  The details of the setup for novel task detection

In the experiments on novel task detection in Sec. 4.1, the number of in-distribution tasks (from the Original domain) in the test set is 4000 (1000 per task cluster) and the number of novel tasks (from the Blur and Pencil domains) in the test set is 8000 (4000 for the Blur and 4000 for the Pencil).

## D  Appendix for Experimental Results

### D.1  Analysis of $\sigma$

| Setting of $2\sigma^{2d}$ | Bird | Texture | Aircraft | Fungi |
|---|---|---|---|---|
| 0.1 | 69.33 | 46.92 | 75.20 | 50.78 |
| 0.5 | 70.00 | **47.98** | 75.38 | **52.38** |
| 1.0 (Ours) | **70.12** | 47.76 | **75.52** | 52.06 |
| 10.0 | 69.4 | 47.28 | 75.32 | 51.5 |

Table 4: Analysis of different $\sigma$

Tabel 4 reports the effect of different $\sigma$ on the classification performance (5-way-1-shot classification on Multi-Plain dataset). As shown in the table, although the too low or too high setting of this hyper-parameter will hurt the performance, in general the model is robust toward the setting of $\sigma$.

## D.2 Analysis of $\bar{\sigma}$

| Setting of $\bar{\sigma}$ | Bird | Texture | Aircraft | Fungi |
|---|---|---|---|---|
| 0.05 | 69.78 | **48.36** | 74.36 | 51.34 |
| 0.1(Ours) | **70.12** | 47.76 | **75.52** | **52.06** |
| 0.2 | 70.02 | 47.50 | 75.30 | 51.74 |
| 0.5 | 69.02 | 46.66 | 74.46 | 51.00 |

Table 5: Analysis of different $\bar{\sigma}$

Tabel 5 summarizes how different settings of $\bar{\sigma}$ influence the classification performance (5-way-1-shot classification on Plain-Multi dataset). In general, different settings of $\bar{\sigma}$ will influence the model performance at a marginal level, indicating our model's robustness toward this hyper-parameter.

## D.3 Impact of GMM component number

| Number of components $r$ | 2 | 4 | 8 | 16 | 32 |
|---|---|---|---|---|---|
| Silhouette score | 47.70 | 57.61 | 12.76 | 7.81 | 6.19 |

Table 6: Analysis on the number of mixture components

Different choices of the number of mixture components does not significantly influence the model classification performance. However, the clustering quality may vary due to the different numbers of components. Here, we report the Silhouette score [38, 39] on Plain-Multi dataset *w.r.t.* the number in Table 6. From Table 6, we can see that selecting a component number close to the ground-truth component number of the distribution can benefit the clustering quality.

## D.4 Classification performance of the ablation variants

| Ablation Variants | Bird | Texture | Aircraft | Fungi |
|---|---|---|---|---|
| HTGM w/o GMM | 68.86 | **48.00** | **75.74** | **52.28** |
| HTGM-Gaussian | 69.52 | 47.3 | 75.38 | 51.34 |
| HTGM | **70.12** | 47.76 | 75.52 | 52.06 |

Table 7: Ablation study of different variants of our proposed method.

We summarize the classification performance of the two Ablation Variants HTGM w/o GMM and HTGM-Gaussian in Table 7. As we can see, our unique designs improve the novel task detection performance without significantly decreasing the classification performance.

## D.5 Ablation analysis of optimization-based methods

| Setting | Model | Bird | Texture | Aircraft | Fungi | Average |
|---|---|---|---|---|---|---|
| | ANIL-MAML | 62.64±0.90 | 43.86±0.78 | 70.03±0.85 | 48.34±0.89 | 56.22 |
| 5-way-1-shot | ANIL-HSML | 64.33±0.87 | 43.77±0.79 | 69.71±0.84 | 47.75±0.89 | 56.39 |
| | ANIL-ARML | 65.98±0.87 | 43.57±0.78 | 70.28±0.84 | 48.48±0.92 | 57.08 |
| | HTGM (ours) | **70.12±1.28** | **47.76±1.49** | **75.52±1.24** | **52.06±1.41** | **61.37** |
| | ANIL-MAML | 74.38±0.73 | 55.36±0.74 | 79.78±0.63 | 59.57±0.79 | 67.27 |
| 5-way-5-shot | ANIL-HSML | 78.18±0.71 | 57.70±0.75 | 81.32±0.62 | 59.83±0.81 | 69.26 |
| | ANIL-ARML | 78.79±0.71 | 57.61±0.73 | 81.86±0.59 | 60.19±0.81 | 69.61 |
| | HTGM (ours) | **82.27±0.74** | **60.67±0.78** | **88.48±0.52** | **65.70±0.79** | **74.28** |

Table 8: More results (accuracy±95% confidence) of the optimization-based methods.

We selected the two best performed optimization-based baselines HSML and ARML, and the widely used method MAML for this ablation analysis. Table 8 summarizes the performance of MAML, HSML and ARML trained in ANIL method [36], *i.e.*, we pre-trained the ResNet-12 by ProtoNet, froze the encoder, and fine-tuned the last fully-connected layers using MAML, HSML and ARML on Plain-Multi dataset. From Table 8, the performance of ANIL-MAML is better than MAML in Table 1, similar to the observation in [36], indicating the effectiveness of ANIL method. However, ANIL-HSML and ANIL-ARML perform similarly to ANIL-MAML, losing their superiority of modeling the mixture distribution of tasks achieved when implemented without ANIL as in Table 1 (up to 5.6% average improvement). This is because the clustering layers in HSML and the graph layers in ARML both affect the embeddings learned through backpropagation, *i.e.*, they were designed for joint training with the encoder. When the encoder is frozen, they cannot work properly. For this reason, to be consistent with the existing research [52, 53] that demonstrated the difference between HSML/ARML and MAML, we used their original designs in Sec. 4. Meanwhile, we observed the proposed HTGM outperforms MAML, HSML, and ARML trained in ANIL method, this is because MAML cannot model the mixture distribution of tasks, while HSML and ARML cannot work properly when trained in ANIL method.

### D.6 More results on the Mini-ImageNet dataset

| Model | 5-way-1-shot | 5-way-5-shot |
|---|---|---|
| ProtoNet-Aug | 59.40±0.93 | **74.68±0.45** |
| HTGM (ours) | **61.80±0.95** | 74.55±0.45 |

Table 9: Comparison of the proposed method and ProtoNet-Aug on the Mini-ImageNet dataset.

In the case when the task distribution is not a mixture, our model would degenerate to and perform similarly to the general metric-based meta-learning methods, *e.g.*, ProtoNet, which only considers a uni-component distribution. To confirm this, we added an experiment that compares our model with ProtoNet-Aug on Mini-ImageNet [47], which does not have the same explicit mixture distributions as in the Plain-Multi and Art-Multi datasets in Section 4. The results are summarized in Table 9. From the table, we observe our method performs comparably to ProtoNet, which validates the aforementioned guess. Meanwhile, together with the results in Table 1 and Table 2, the proposed method could be considered as a generalization of the metric-based methods to the mixture of task distributions.

### D.7 Task Embedding Visualization.

We provide visualizations of the task embeddings learnt by HTGM. Fig 4 is the visualization of the task embeddings for Few-Shot Classification on Plain-Multi dataset (Sec. 4.1) via t-SNE [45]. Every yellow point corresponds to a task sampled from Aircraft. Every blue point corresponds to a task sampled from Texture. Every red point corresponds to a task sampled from Fungi. Every green point corresponds to a task sampled from Bird. As we can see, in general, different classes of tasks are well clustered in the task embedding space, indicating that HTGM learnt task embeddings that capture the difference of tasks from different groups.

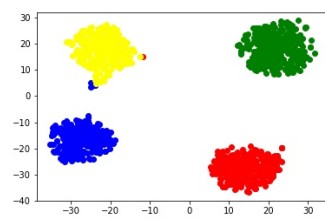

Figure 4: Clusters of task embeddings on Plain-Multi.

### D.8 Computational Cost

We report the cost of ProtoNet-Aug, NCA, FEATS and our model because they use the same encoder architecture. We evaluated and trained all of the models on RTX 6000 GPU with 24 GB memory.

Training Time: According to the training logs, the training of the ProtoNet-Aug took 10 hours, the training of NCA took 6.5 hours, the training of FEATS took 10.5 hours, and the training of the

proposed model HTGM took 13 hours. Please note that our algorithm and FEATS require pre-training the encoder with ProtoNet-Aug. The 10.5 and 13 hours include the 10 hours pre-training phase. The main cost of our model is not from the energy function, because we have reduced its partition function to a constant using Eq. (6) in Theorem 3.1, whose training cost is negligible.

The higher cost is because (1) our model needs to jointly learn a GMM model in every EM step and (2) jointly learning the generative model and classification model takes more learning steps to converge. Given the pre-trained ProtoNet-Aug encoder, the FEATS took about 3000 steps to converge, the proposed model took about 10000 steps to converge. In other words, the EM training algorithm takes more computational overhead. However, the advantage comes with the training cost is the better classification and novel task detection performance.

Test Time: Because we approximated the partition function of the energy function with a constant upper bound, we almost added zero computational cost to the model inference. The test time of NCA, FEATS, ProtoNet, and our model are all around 85-95 seconds for 1000 tasks.

