# OpenReview forum: "Hierarchical Gaussian Mixture based Task Generative Model for Robust Meta-Learning"
_NeurIPS.cc/2023/Conference — NeurIPS 2023 poster_

### Official Review · Reviewer_dA8b · 2023-07-04

**Soundness:** 3 good
**Presentation:** 3 good
**Contribution:** 3 good
**Rating:** 6
**Confidence:** 3

**Summary:**

The authors propose a novel probabilistic meta-learning algorithm called HTGM
which models the full hierarchical sampling procedure in few-shot classification
in order to perform prediction and novel task detection. To learn the parameters
they employ the EM procedure together with variational inference. Experimental
results show that they perform best among competitors in few-shot classification
prediction and novel task detection.

**Strengths:**

- Originality & significance: The authors propose a probabilistic model of the full few-shot
  classification process which allows for both adaptation (prediction) and
  novelty detection. Additionally it can be added on top of other metric-learning few-shot learning algorithms out of the box.
- Quality: The probabilistic model seem reasonable and they discuss problems and
  solutions of design choices made. Experiments for both prediction and novelty-detection is convincing (but see the weakness section).
- Clarity: The paper is well-written in general.


**Weaknesses:**

- Partition function: The authors show that the partition function is upper bounded and show asymptotic tightness of this bound for the minimal distance $D_{hl}$, but I'm still not fully convinced by this and would like to know more how this impacts modelling and "what we lose" in disregarding the partition function.
- Probabilistic modelling and "tricks": The pipeline has several tricks which may help performance but weakens the interpretation of the model as inferring a correct probabilistic distribution.
- Non-standard multi-domain experiments: The datasets of Plain-Multi and
  Art-Multi are non-standard. Ideally additional experiments on for example
  meta-dataset would be beneficial.
- Typos:
  1. Line 172: I think $\mu_k^c$ should be $\mu_1^c$ and $\mu_N^c$?
  2. Line 314: comapred -> compared
  3. Line 341: they even don't fit -> they don't even fit
  4. Table 2: I think the second entry in the "Setting" column should be "5-way, 5-shot"


**Questions:**

# Partition Function
Could you explain what the potential limitations of disregarding the partition
function is? Intuitively it feels like the partition function would have a
regularizatory effect and disregarding it may lead to some overfitting. Would be
interested to hear what the authors think the effect of being able to optimizing
this part of the objective would be.

# Tricks
- Tying $\theta$ and $\phi$: Is there are reason for letting the recognition
  network $q$ be equal to $f$? Is the tying of the parameters done due to
  convenience and that it empirically was found to work well or is there some other reason?
- Negative sampling and partition function: Show empirical results that the
  quantity $D\_{hl}$ goes to infinity over the course of training.
- Model adaptation: Is it possible to put the adaptation step on a probabilistic
  footing? Right now it seems like it's pretty ad-hoc, can we for example trust
  $p(y'\_i = j' | x\_i')$ more as a probabilistic model compared to using any
  standard (non-probabilistic) metric learning few-shot algorithm which produces
  an embedding $f_{\theta}(x)$? One way to show this would be to show that $p$
  here is more well-callibrated as opposed to using the backbone trained by
  proto-net or some other algorithm.

# Non-standard experimental dataset
Meta-dataset is the standard modern community benchmark for few-shot
multi-domain classification. I'd be interested to see how the algorithm fares on
this benchmark since their probabilistic model explicitly models the
multi-domain aspect of few-shot learning for which the meta-dataset seems like a
good fit.


**Limitations:**

No, certain limitations should be expanded upon (see Questions). Societal impact is discussed correctly.

---

> ### Author Rebuttal · Authors · 2023-08-10
>
> Thank you so much for the constructive feedback. Here are our responses.
>
> Q1. Limitations of disregarding the partition function
>
> Response: The main issue of replacing the partition function with its upper bound is that it increases the noise in the inferred class mean. Note that the entropy of distribution $\pi(\mu_k^c|v_{\tau},w)$ ($\mu_k^c$ is a class mean) in Eq. (2) is that:
> $$H(\pi)=-\int\pi(\mu_k^c|v_{\tau},w)\log\pi(\mu_k^c|v_{\tau},w)d\mu_k^c =\log Z+\bar{E_w}$$
> where $\bar{E_w}$ is the average energy. Therefore replacing $Z$ with its upper bound increases the entropy of $\pi(\mu_k^c|v_{\tau},w)$, leading to more noise in $\mu_k^c$. This leads to a decreasing accuracy of meta-learning. To alleviate this issue, we proposed to include an negative sampling term as in Eq. (4) to increase the class mean distance to reduce the influence of the noise
>
> Q2.1 Tying parameters.
>
> Response: The purpose of the inference network $\phi$ in $q_{\phi}$ is to infer $v_{\tau}$ from the support set $D_{\tau}^{s}$ in the embedding space, where the input $D_{\tau}^{s}$ is the same as an encoding network $f$. Thus we built the inference network $\phi$ upon $f$ (with two aggregation functions) to minimize parameter use for avoiding overcomplicated model. As in Fig 1, the two aggregation functions first aggregates the sample embedings in the support set $D_{\tau}^{s}$ to class embeddings, then aggregates the class embeddings to the task embedding $v_{\tau}$. This is similar to VAE where the inference network is an encoder, but in our case, the inference network coincides with the encoder (a.k.a., the base model) in the metric-based meta-learning.
>
> On the other hand, we think it is unnecessary to define two encoders, one for metric-based meta-learning, i.e., classifying the sample embedings, another for the inference network, i.e., infer task embeddings. This is because the task embedding is inferred from the class embeddings, thus classifying sample embeddings is inevitable in the inference network, making the former encoder a duplicate. Hence, it is intuitive to build $\phi$ in $q_{\phi}$ upon $f$.
>
> However, it is possible to add more parameters on $f$ to form $\phi$, such as using parameteric aggregation functions than non-parametric ones, as the framework is flexible. In this work, we used the simplest model structure as it was found suffiently effective to demonstrate the usefulness of the proposed model.
>
> Q2.2 Empirical results of quantity $D_{hl}$.
>
> Response: We tested the average $D_{hl}$ of our model on different training steps. And with $D_{hl}$, we can estimate the error ratio of our approximation to the partition function. This is because in our model, the error ratio $\frac{N\sqrt{2^{d-1}\pi^d}-Z}{N\sqrt{2^{d-1}\pi^d}}$ is bounded by $er(D_{hl})=\frac{\gamma(\frac{d}{2},D_{hl}^2/4)}{(\frac{d}{2}-1)!}$. We calculate them for each average $D_{hl}$ and include them in the Fig. 2 in the common rebuttal. As we can see, as training goes by, the error ratio of our approximation monotonically decreases. At step 7000, where the error ratio is 0.25, the model acquires the best validation accuracy, which means that 0.25 is enough for the model to handle the tasks.
>
> Q2.3 Model adaptation with probablistic explanation.
>
> Response: Yes, this model adaptation has a probablistic explanation consisting of two steps.
>
> First we learn class means on support set: When a meta-testing episode comes, for each class $y_j$, we first encode its support samples to acquire its support sample embeddings
> $e^{(j)}_1,e^{(j)}_2,...,e^{(j)}_k$.
>
> Then we infer the task embedding $v_{\tau}$ via $q_{\Phi}(v_{\tau}|D^s_{\tau})$ (in line 208). After that, we freeze other parameters and learn class embedding $\bar{\mu}_j^c$ for each class $y_j$
> via maximum likelihood estimation. According to line 184-187, the likelihood function that we observe the support set of each class $y_j$ should be
> $$L(\bar{\mu}_j^c)=\log p({y}_j|v{\tau})\prod_1^k p({e}{i}^{(j)}|{y}j))=\log \pi(\bar{\mu}^ck|v{\tau},w)\prod_1^k\mathcal{N}(e^{(j)}_i|\bar{\mu}^c,\Sigma)$$
>
> i.e. $p(y_j|v_{\tau})$ is the likelihood of step (a) in line 184, and $\prod_1^k p(e^{(j)}i|yj))$ is the likelihood of step (b) in line 185-186 (the inner loop). Our theoretic analysis in the common rebuttal shows the adaptation step $\bar{\mu}_j=\alpha{\mu}^c + (1-\alpha)W{l}v{\tau}$
> maximizes the above likelihood function.
>
> However, our empirical evaluation shows that sometimes the emperically optimal value of $\alpha$ might be different from theoretically optimal value $\frac{k}{k+2\sigma^{2d}}$ (See the common rebuttal). We suggests that it might be because the noise in partition function approximation (see the answer to Q1). Therefore, we treat $\alpha$ as a hyperparameter to fineunte on validation set.
>
> Second, we classify the query sample $x'$: We first compute $p(y'=j'|x')$ i.e. the posterior distribution of the query sample's label $y'$ conditioned on the sample. Then we select the label with the highest probability. Given the learnt class means, we compute the conditional distribution of label:
> \begin{equation}
>     p(y'_i=j'|x'_i) = \frac{p(y'_i=j',x'_i)}{\sum_j^Np(y'_i=j,x'_i)} =\frac{p(x'_i|y'_i=j')p(y'_i=j')}{\sum_j^Np(x'_i|y'_i=j)p(y'_i=j)}
> \end{equation}
> During the meta-testing stage, we should equally treat each class, i.e. assuming $p(y'_i=1)=...=p(y'_i=N)$. So we have:
> \begin{equation}
>    p(y'_i=j'|x'_i) = \frac{\exp(||f{\theta}(x'i)-\bar{\mu}{j'}^c||}{\sum_j^N\exp(||f{\theta}(x'i)-\bar{\mu}j^c||}
> \end{equation}
>
> Q3. Non-standard experiment dataset
>
> Response: We used the current datasets in our experiments because they were regarded as benchmarks and used by other mixture-distribution based meta-learning works (e.g., [47]). Among them, the Plain-Multi dataset consists of four datasets that also exists in Meta-Dataset. The main difference is that this benchmark does not include Mini-ImageNet. Thus, we provided an experiment on Mini-ImageNet dataset in our Appendix D.6

---

> > ### Comment · Reviewer_dA8b · 2023-08-14
> > **Thank you**
> >
> > I appreciate the detailed response. I believe Q1 and Q2 to be successfully answered but I am unsure if the method will work on harder datasets such as meta-dataset. I have decided to keep my score as is as I still think the work has merit and should be accepted.

---

> > > ### Author Response · Authors · 2023-08-19
> > >
> > > Thank you so much! We will include more discussions into our draft if it is accepted.
> > >
> > > Best regards,
> > >
> > > Authors of this paper.

---

### Official Review · Reviewer_KXFG · 2023-07-05

**Soundness:** 2 fair
**Presentation:** 3 good
**Contribution:** 2 fair
**Rating:** 5
**Confidence:** 4

**Summary:**

This paper studies the task distribution in meta learning and proposes modeling the task in multimodal distributions. To enable efficient modelling and inference, the author develops the hierarchical Gaussian mixture task generative mode and optimizes the meta learning model in expectation maximization. The performance of the method is verified in few-shot image classification tasks.

**Strengths:**

1. This paper is written clearly, and the modelling of task distribution has great significance in the literature.
2. It shows improved performance in adopted benchmarks over existing metrics based methods.

**Weaknesses:**

(1) Probabilistic relations and graphical models.

It is necessary to explain why the generative process Eq. (1) is valid in metrics-based meta learning. In practice, meta learning seldom considers the sample embeddings and the task embedding is of interest. Hence, the conditional distribution $\ln\int p(D_{\tau}^{q}| v_{\tau})p(v_{\tau}\vert D_{\tau}^{s})dv_{\tau}$ is mostly commonly used as the objective to maximize. However, in Eq.(1), the conditional dependencies on the support dataset are neglected. My suggestion is that this needs to be reflected in Eq.(1) directly.

(2) Model optimization.

In Line201-202, it says, “the query set is not included because it is unavailable during model testing”. This does not hold in Bayesian meta learning. Like that in conditional variational autoencoder or neural processes, the $q_{\phi}(v_{\tau}|D_{\tau}^{s})$ is called the approximate prior and the approximate posterior $q_{\phi}(v_{\tau}|D_{\tau}^{s}, D_{\tau}^{q})$ is used in meta training for more effective inference (though the query set is unavailable in meta testing). Hence, I doubt the way of inference in this work since the conditional prior is most important for meta learning probabilistic models. Especially, the way to infer $e_i$ throughout the manuscript, e.g., $p(e_i|y_i)$ in Eq.(3)-(7), the one-hot information can be quite limited. However, the input $x_i$ is not included in the inference.

(3) Mixture of task distributions.

The task embeddings lie in a mixture of Gaussian manifold is a reasonable assumption. For the task2vec work, the embedding of the task is visualized and analyzed a bit. In this work, the focus is on task distribution, so it is necessary to include this part in experimental analysis, for example, to check the grouping of latent structures in the task space.

**Questions:**

(1) Would you please provide the detailed probabilistic graphical model to explain the relationship between the support, the query, the sample-wise embedding and etc?

(2) Since the experiments were performed on typical benchmarks, would you explain the novel tasks a bit and how to define the discovery of novel tasks?

(3) Are task distributions the same for the meta training and meta testing, or do out-of-distribution cases exist?

**Limitations:**

See the weakness section.

---

> ### Author Rebuttal · Authors · 2023-08-10
>
> Thank you for helping us refine the work. Before answering the questions, please allow us to clarify our problem setting.
> In this work, we aim to develop a robust meta-learning model for mixture task distribution such that when a meta-testing task (e.g. few-shot classification) comes, the model can:
> 1. identify whether the task comes from the same mixture distribution of meta-training set (i.e., a known task), or a different distribution (i.e., a novel task).
> 2. if the meta-testing task is a known task, adapt the model to this task with the task-level information for boosted performance.
> The motivation is that in areas requiring robust decisions, such as healthcare, where the accuracy drop on novel tasks is inevitable, a robust and ethically safe choice is to raise an alarm to the users (e.g. doctors) for diagnosis and decision-making.
>
> Response to Weakness (1): Please let us validate our design from two prespectives.
>
> 1. The relation to the commonly used $\ln\int p(D^q_{\tau}|v_{tau})p(v_{tau}|D^s_{\tau})d_{v_{\tau}}$
> As in the aforementioned problem setting, our work is trying to handle both novel task detection and meta-learning. The commonly used $\ln\int p(D^q_{\tau}|v_{\tau})p(v_{\tau}|D^s_{\tau})d_{v_{\tau}}=p(D^q_{\tau}|D^s_{\tau})$ in existing works only considers the how to infer the query set information (e.g. labels) given support set (i.e. meta-learning). Therefore, we propose to handle novel task detection by modeling the likelihood of a task $p(D^q_{\tau},D^s_{\tau})$ (Eq. (1)), so that we can use a low likelihood score to detect novel tasks in meta-testing. Also, note $p(D^q_{\tau},D^s_{\tau})=p(D^q_{\tau}|D^s_{\tau})p(D^s_{\tau})$. Thus, our training objective function also contained the information of the commonly used likelihood function.
>
> 2. The relation between $p(D_{\tau}|v_{\tau})$ and the sample embeddings. In our work, we modeled $p(D^q_{\tau}|v_{\tau})$ as (Eq. 1):
> $$\log p(D_{\tau}=\\{(x_i,y_i)\\}|v_{\tau})=\sum_i^n \log p(x_i|y_i)p(y_i|v_{\tau})$$
> This is because we assume the samples in the task are i.i.d., so we can factorize the distribution to each sample's probability density. We modeled the sample probability density with their sample embeddings and class mean:
> $$p(x_i|y_i) = \frac{\exp(||f_{\theta}(x_i),y_i||^2_2)}{\sigma^2I} =\mathcal{N}(e_i|\mu_{y_i}^c,\sigma^2I)$$
> This operation is common in representation learning. For example, in [1], the authors applied similar definition, whose difference from us is that they applied inner product rather than Euclidean distance. In Eq. 1 of our draft, we directly wrote the $p(x_i|y_i)$ as $p(e_i|y_i)$ to emphasize that this probability can be efficiently computed in embedding space.
>
> Response to Weakness (2): The difference between our model and the example in the review (conditional VAE and Neural Process) is as follows. In our case, during the meta-testing, the infered task embedding $v_{\tau}$ is fed to the learnt GMM model to compute the prior likelihood $p(v_{\tau})$. The likelihood value is used to detect novel tasks. For effective detection, we need to ensure the infered $v_{\tau}$ of in-distribution tasks in meta-testing to have the same distribution as the infered $v_{\tau}$ of meta-training. In the meta-testing stage, since the query set is not available for novel task detection, we can only infer the $v_{\tau}$ via $q(v_{\tau}|D^s_{\tau})$. If we use $q(v_{\tau}|D^s_{\tau}, D^q_{\tau})$ in meta-training for more powerful inference, there could be a distribution discrepancy between the in-distribution tasks $v_{\tau}$ in the meta-training set and meta-testing set. Specifically, the variance of $q(v_{\tau}|D^s_{\tau})$ may be larger than $q(v_{\tau}|D^s_{\tau}, D^q_{\tau})$ as the former is estimated on fewer samples.
>
> Response to Weakness (3): We included the visualization of task embedding clusters in the common rebuttal PDF file.
>
> Q1.Detailed probabilistic graphical model?
>
> We have drawn the probalistic graph for both generation and inference in the commom rebuttal PDF file (Fig. 3).
>
> Q2.Explain the novel tasks and how to define the discovery of novel tasks?
>
> In typical benchmarks of meta-learning, we evaluate models by sampling a meta-testing dataset, where each data instance in this dataset is a task containing a support set and a query set that are both sampled from the same distribution of the meta-training dataset. The accuracy of meta-learning algorithms usually refers to the average performance of the model on all meta-testing tasks. In our work, a novel task, as we explained in line 22-25 and line 312-316, is a task whose support samples and a query samples are not drawn from the same distribution of the meta-training dataset. In our experiment of novel task detection, as we explained in line 312-316, we applied two typical benchmarks, Art-Muli and Plain-Multi, whose data samples (images) are drawn from different distributions. We then trained meta-learning models only on the Plain-Multi dataset. After that, during meta-testing stage, we mix the meta-testing tasks drawn from Art-Muli and Plain-Multi and then evaluate the models' ability to detect the meta-testing tasks drawn from Art-Muli (i.e. novel tasks).
>
> Q3. Do out-of-distribution cases exist?
>
> If we understand it correctly, this question is about the settings in our two experiments.
> In the few-shot classification experiment, since some baselines cannot handle out-of-distribution cases, for fair comparison, we only evaluated the accuracy of our model and baseline models on in-distribution tasks. In this case, out-of-distribution tasks do not exist in the meta-testing set. In the novel task detection experiment, we compared our model with the baselines that can detect out-of-distribution tasks. In this case, we mixed the in-distribution tasks and out-of-distribution tasks and evaluated the ability of the models in detecting the out-of-distribution tasks.
>
> [1] Jian Tang, et al. LINE: Large-scale Information Network Embedding. WWW 2015

---

> ### Author Response · Authors · 2023-08-18
>
> Dear Reviewer KXFG,
>
> Thank you so much for providing the constructive feedbacks for our work. We have tried our best to address the questions in your review. We will sincerely appreciate if we could hear from you on whether there are any remaining questions or concerns, so that we can take this opportunity to further clarify them and improve our work. Thanks a lot!
>
> Best regards,
>
> Authors of this paper

---

> > ### Comment · Reviewer_KXFG · 2023-08-19
> >
> > Thanks for the response. Most of conerns are addressed. Some can be future exploration, e.g., investigating the inference method like that in conditional prior like neural processes. Also, it is encouraged to release the code later for better comprehension. I've increased my score.

---

> > > ### Author Response · Authors · 2023-08-20
> > >
> > > Dear Reviewer KXFG,
> > >
> > > We sincerely appreciate your review and comments! If this paper is accepted, we will discuss the future explorations accordingly and opensource our code.
> > >
> > > Best regards,
> > >
> > > Authors of this paper.

---

### Official Review · Reviewer_XMuD · 2023-07-05

**Soundness:** 3 good
**Presentation:** 3 good
**Contribution:** 2 fair
**Rating:** 5
**Confidence:** 4

**Summary:**

This paper proposes a Hierarchical Gaussian Mixture method as a means to parameterize the task generation process in meta-learning. The authors suggest that their proposed model can effectively fit a mixture of task distributions and evaluate the scoring of testing tasks. The effectiveness of the proposed methods is demonstrated through experiments conducted on multiple datasets.

**Strengths:**

* This paper is well-written and easy to follow


* It is interesting that the proposed method can meta-learn both a mixture of task distributions and detect novelty in testing tasks.

**Weaknesses:**

*  In this paper, the authors present an application of the Hierarchical Gaussian Mixture model to the meta-learning setting. While the paper provides ideas into utilizing this model for meta-learning, the method itself is not novel and has been extensively studied within the machine learning community.

* The paper's theoretical analysis appears limited and would benefit from a more in-depth analysis. Specifically, it is crucial to provide a comprehensive analysis of the generalization bound associated with the proposed methods.

**Questions:**

N/A

**Limitations:**

The authors have clearly addressed the limitations and potential negative societal impact of their work

---

> ### Author Rebuttal · Authors · 2023-08-10
>
> Thank you for your review. Here are our feedbacks to the review:
>
> Response to Weakness 1: In this paper, the authors present an application of the Hierarchical Gaussian Mixture model to the meta-learning setting. While the paper provides ideas into utilizing this model for meta-learning, the method itself is not novel and has been extensively studied within the machine learning community.
>
> Response: Please let us clarify the difference between our method and the existing works on Hierarchical Gaussian Mixture (HGM) model (which was also discussed in the last paragraph in Section 2 and Appendix B.2).
>
> To the best of our knowledge, the Hierarchical Gaussian Mixture (HGM) model has appeared in some traditional works [8,30,3] for hierarchical clustering by applying Gaussian Mixture model agglomeratively or divisively on the input samples. They are unsupervised methods that infer clusters of samples, but do not pre-train embedding models (or parameter initializations) that could be fine-tuned for the adaptation to new tasks in meta-learning. Therefore, these methods are remarkably different from meta-learning methods, and we think it is a non-trivial problem to adapt the concept of HGM to solve the meta-learning problem. To this end, we need to (1) identify the motivation; and (2) solve the new technical challenges.
>
> For (1), we found the hierarchical structure of mixture distributions naturally appears when we want to model the generative process of tasks from a mixture of distributions, where each task contains another mixture distribution of classes (as suggested by Eq. 1). In other words, the motivating point of our method is more on meta-learning than HGM. However, we think drawing such a connection between meta-learning and HGM is a novel contribution.
>
> For (2), our method is different from traditional HGM in (a) its generative process of tasks (Sec. 3.1), which is a theoretical extension of the widely used empirical process of generating tasks in meta-learning; (b) its Gibbs-style task-conditional distribution (Eq. 2) for fitting uniformly sampled classes; (c) the metric-based end-to-end meta-learning framework (Fig. 1) (note the traditional HGM is not for learning embeddings); (d) the non-trivial derivation of the optimization algorithm in Sect. 3.2 and Alg. 1; and (e) the novel model adaptation process in Sec. 3.3. Solving the technical challenges in the new generative model is another novel contribution of the proposed method.
>
> As such, we think our work is a new meta-learning method with a mixture task generative model.
>
> Response to Weakness 2: The paper's theoretical analysis appears limited and would benefit from a more in-depth analysis. Specifically, it is crucial to provide a comprehensive analysis of the generalization bound associated with the proposed methods.
>
> Response: In this paper, we mainly have provided theoretic contributions from the following perspectives:
>
> 1. We extended the widely used empirical process of generating a task to a theoretical process specified by a hierarchy of Gaussian mixture (GM) distributions. HTGM generates a task embedding from a task-level GM, and uses it to define the task-conditioned mixture probabilities for a class-level GM, from which the data samples are drawn for instantiating the generated task.
>
> 2. To address the challenge of computing the partition function in the Gibbs distribution in the aforementioned class-level GM (in Eq. (2)), we propose to replace it with a theoretically justified upper bound (in Theorem 3.1).
>
> 3. In the global rebuttal (and the uploaded PDF file), we additionally provide a theoretic result indicating that our simple model adaptation strategy (in Section 3.3) is actually an Maximum Likelihood Estimation of the class means.
>
> We think the above three contributions can help sufficiently justify the novel design in our proposed model. Meanwhile, it is noteworthy that in this work, we are actually handling the generalization gap by detecting novel tasks (that is out-of-distribution) with low likelihood and raises alerts. Therefore, the generalization bound is only meaningful for the in-distribution tasks. As for the general in-distribution cases of non-convex models trained with SGD, [1] gives a good generalization bound, ensuring the effectiveness of the general framework applied in our model. Therefore, in this work, we mainly focus on theoretically justifying the aforementioned unique designs.
>
> [1] Kuzborskij, Ilja, and Christoph Lampert. "Data-dependent stability of stochastic gradient descent." International Conference on Machine Learning. PMLR, 2018.

---

> ### Author Response · Authors · 2023-08-18
>
> Dear Reviewer XMuD,
>
> We are very thankful to your review on our paper. We have provided our response to the questions and concerns in your review. We will sincerely appreciate if you could let us know if there are any remaining questions or concerns so that we can take this opportunity to refine our work? Thanks a lot!
>
> Best regards,
>
> Authors of this paper

---

> > ### Comment · Reviewer_XMuD · 2023-08-19
> > **Thanks for your response**
> >
> > Dear authors,
> >
> > Thanks for your response. I read the rebuttal and addressed some of my concerns, I increased my score to 5.

---

> > > ### Author Response · Authors · 2023-08-20
> > > **Thank you for your feedback!**
> > >
> > > Dear Reviewer XMuD,
> > >
> > > We are really thankful for your review and feedback!
> > >
> > > Best regards,
> > >
> > > Authors of this paper.

---

### Official Review · Reviewer_k7fv · 2023-07-06

**Soundness:** 3 good
**Presentation:** 3 good
**Contribution:** 3 good
**Rating:** 6
**Confidence:** 4

**Summary:**

In realistic scenario, training tasks and test tasks may come from different distributions. However, most existing meat-learning methods do not take it into account. Even if some do, they do not jointly handle the detection of novel tasks.

The authors propose a metric-based meta learning model that handles both the mixture of task distributions and detection of novel tasks. The effectiveness of the proposed method compared to previous methods is verified on several benchmark datasets.

**Strengths:**

1. It is novel to approximate the distribution for tasks. In particular, it aligns with realistic scenarios by modeling it as a mixture of distributions.
2. The paper is well-written; it clearly explains its gaol with mathematical support, and notations are well-described and consistent. Figures are also helpful to understand the proposed idea.
3. Although Eq.(3) still cannot be computed due to the normalizing constant in $p_\omega(y_i|v_\tau)$, the authors provided the reasonable workaround in Theorem 3.1, which is also an interesting contribution.

**Weaknesses:**

1. Although $l_{HTGM}$ is well-motivated and clear, $l_{neg}$ is somewhat heuristic.
2. If my understanding is correct, a large part of computational overhead compared to baseline methods e.g. ProtoNet comes from E-step and GMM. Especially, conducting GMM at each epoch can be quite expensive. But the analysis for computational cost is not provided. Can the authors provide the computational overhead (e.g. in wall-clock time or flops) of the proposed method (and time complexity if possible)?
3. To demonstrate that the effectiveness of the method on cross domain, the experiments on two datasets the authors used are not sufficient, especially, if we want to validate whether it works in practice or not. It could have been much more plausible if experiments were conducted on a larger scale dataset such as Meta-Dataset.

**Questions:**

1. I am a bit confused with Eq.(4)-(5). Here, the objective of the additional negative sampling loss is to make the different class means disperse. To this end, Eq.(4) is used. But, if we minimize $l_{neg}$, wouldn’t it force $e_{j}$ and $\mu$ to be close to each other? Please correct me if I am wrong.
2. Could the authors elaborate how to interpret the Figure 2? In my understanding, the more distant two tasks are, the better a model is to distinguish two task groups. If so, measuring the distance between two normalized likelihood histogram would be helpful (treating them as probability distributions if necessary). Intuitively, the distance may be roughly estimated as the difference between the red and blue regions. I do not see the significant difference between the models (except HSML).
3. Is there any particular reason why EM is used over VI?

**Limitations:**

Please refer to the weakness and questions above.

---

> ### Author Rebuttal · Authors · 2023-08-10
>
> Thank you so much for the constructive feedback. We sincerely appreciate your valuable suggestions and questions. The following are our responses.
>
> Q1.I am a bit confused with Eq.(4)-(5). Here, the objective of the additional negative sampling loss is to make the different class means disperse. To this end, Eq.(4) is used. But, if we minimize, wouldn’t it force and to be close to each other? Please correct me if I am wrong.
>
> Response: Thank you for the question. In our draft, we are maximizing our objective function (the lower bound of the likelihood in Eq. (3)). Thus, when we add the negative sampling term to the objective function, we are jointly maximizing it together with Eq. (3) (not minimizing it), so that the optimization will force the class means to disperse.
>
> Q2. Could the authors elaborate how to interpret the Figure 2? In my understanding, the more distant two tasks are, the better a model is to distinguish two task groups. If so, measuring the distance between two normalized likelihood histogram would be helpful (treating them as probability distributions if necessary). Intuitively, the distance may be roughly estimated as the difference between the red and blue regions. I do not see the significant difference between the models (except HSML).
>
> Response: Your understanding is right. To better interpret the figures, we calculated the ratio of non-overlap-area/total-area in the four figures. The higher this value is, the more distant the two distributions are. Here is the result:
>
> HSML:0.1379;
>
> MetaOpt:0.4952;
>
> ProtoNet-Aug: 0.4806;
>
> HTGM:0.5578;
>
> As we can see, the ratio of non-overlap-area of HTGM is larger, which means that the two distributions are more distant. We will include these results in a revised draft.
>
>
> Q3. Is there any particular reason why EM is used over VI?
>
> Response: If we are understanding correctly, this question is asking why we use EM to train the loss of variational inference instead of using common SGD (e.g. Variational Auto-Encoder uses SGD). The reason is that GMM in our model can not be trivially trained with SGD. This is because GMM has some constraint on its parameters (e.g. its covariated matrix must be semi-positive-definite). Therefore, we apply EM algorithm for VI (variational inference), so that in the E-step, we can use non-SGD algorithm to optimize GMM, and in the M-step, we can optimize other neural network parameters with SGD. Detailed algorithms are included in the Appendix A.4. If using VI incorporated with SGD, the GMM may get invalid parameters.
>
> Q4. (From Weakness 2) Can the authors provide the computational overhead (e.g. in wall-clock time or flops) of the proposed method (and time complexity if possible)?
>
> Response: Yes. Let us report the cost of ProtoNet-Aug, NCA, FEATS and our model because they use the same encoder architecture. We evaluated and trained all of the models on RTX 6000 GPU with 24 GB memory. We will include the time comparison in the Appendix.
>
> Training Time: According to the training logs, the training of the ProtoNet-Aug took 10 hours, the training of NCA took 6.5 hours, the training of FEATS took 10.5 hours, and the training of the proposed model HTGM took 13 hours. Please note that our algorithm and FEATS require pre-training the encoder with ProtoNet-Aug. The 10.5 and 13 hours include the 10 hours pre-training phase. The major cost of our model is not from the energy function, because we have reduced its partition function to a constant using Eq. (6) in Theorem 3.1, whose training cost is negligible.
>
> The higher cost is because (1) our model needs to jointly learn a GMM model in every EM step and (2) jointly learning the generative model and classification model takes more learning steps to converge. Given the pre-trained ProtoNet-Aug encoder, the FEATS took about 3000 steps to converge, the proposed model took about 10000 steps to converge. So we agree with the reviewer that the EM training algorithm takes more computational overhead. However, the advantage comes with the training cost is the better classification and novel task detection performance.
>
> Test Time: Because we approximated the partition function of the energy function with a constant upper bound, we almost added zero computational cost to the model inference. The test time of NCA, FEATS, ProtoNet, and our model are all around 85-95 seconds for 1000 tasks.
>
> Q5. (From Weakness 3) It could have been much more plausible if experiments were conducted on a larger scale dataset such as Meta-Dataset.
>
> Response: Thank you for this suggestion. We used the current datasets in our experiments because they were regarded as benchmarks and used by other mixture-distribution based meta-learning works (e.g., [47]). Among them, the Plain-Multi dataset consists of four datasets that also exists in Meta-Dataset. The main difference is that this benchmark does not include Mini-ImageNet. Thus, we added the following discussion and an experiment on Mini-ImageNet dataset to the Appendix.
>
> "In the case when the task distribution is not a mixture, our model would degenerate to and perform similarly to the general metric-based meta-learning methods, e.g., ProtoNet, which only considers a uni-component distribution. To confirm this, we added an experiment that compares our model with ProtoNet-Aug on Mini-ImageNet, which does not have the same explicit mixture distributions as in the Plain-Multi and Art-Multi datasets in Section 4. The results are summarized in the following table. From the table, we observe our method performs genrally better than ProtoNet-Aug, which validates the aforementioned guess. Meanwhile, together with the results in Table 1 and Table 2, the proposed method could be considered as a generalization of the metric-based methods to the mixture of task distributions"
>
> |Model       | 5-way-1-shot | 5-way-5-shot |
> |------------|--------------|--------------|
> |ProtoNet-Aug|59.40±0.93    |**74.68±0.45**|
> |HTGM        |**61.80±0.95**|74.55±0.45    |

---

> > ### Comment · Reviewer_k7fv · 2023-08-13
> > **Response to the Authors**
> >
> > Thank you for the detailed explanations.
> >
> > It is much clearer, in particular, the comparison in time would be helpful for practitioners  and the response for Q5 is insightful to better understand the relationship between the metric-based model and the proposed model.
> >
> > One last thing I want to ask to the authors is if the authors have a plan to release the code upon acceptance. I believe it would be helpful for the readers to better understand the paper if the code is available.

---

> > > ### Author Response · Authors · 2023-08-13
> > > **Response by authors**
> > >
> > > Thank you so much! Sure, we will open source the code if the paper is accepted. Also, if this paper is accepted, we will include the discussions during the rebuttal in the draft.

---

### Author Rebuttal · Authors · 2023-08-10

Dear Reviewers,

We sincerely appreciate your valuable comments that help us refine our work. If you have more questions or concerns about our response or current draft, please let us know. We are happy to discuss with you. We have uploaded a PDF file that contains figure and addtitional theoretic analysis for reference.

Best regards,

Authors

---

### Decision · Program_Chairs · 2023-09-21

**Decision:**

Accept (poster)

**Comment:**

The four reviews of this paper all leaned towards acceptance but not strongly, with two borderline accept and two weak accept recommendations; none of the reviewers wished to champion the paper.

Several positive aspects were appreciated:
+ One reviewer considered the modeling of tasks as a mixture of distributions as novel (k7fv)
+ The probabilistic model was considered reasonable (dA8b)
+ Meta-learning both the mixture of task distributions and detecting novelty of testing tasks was appreciated (XMuD)
+ The workaround for computing eq. 3 was appreciated (k7fv)
+ Experiments were considered convincing (dA8b)
+ The improved performance over existing methods was appreciated (KXFG)
+ The paper was considered well written (k7fv,KXFG,dA8b) and easy to follow (XMuD)

However, concerns were also raised:
- There was concern of novelty: while the paper applies the hierarchical Gaussian mixture model to meta-learning, the method itself is not novel (XMuD). Authors provided some argumentation towards novelties in a rebuttal.
- There was concern about validity of the generative process (KXFG). Reviewers provided argumentation in a rebuttal.
- Somewhat similarly, there was concern that various tricks of the pipeline might weaken the interpretation that the model infers a correct distribution (dA8b)
- One reviewer was not convinced by the upper bound of the partition function and wished more discussion of what is lost by disregarding the partition function (dA8b); authors provided discussion in a rebuttal.
- There was a desire to check the groupings of tasks in a latent space (KXFG); authors provided an embedding visualization in a rebuttal.
- A deteled graphical model of the relationships between variables such as the support, query, and sample-wise embedding was desired. (KXFG); authors provided a graph drawing in a rebuttal.
- There was criticism of the notion that the query set is not included as it is unavailable during model testing; a reviewer argued that does not hold in Bayesian meta-learning (KXFG)
- The theoretical analysis was considered limited, and an analysis of the generalization bound was desired (XMuD)
- The loss function with negative sampling l_neg was considered somewhat heuristic (k7fv)
- Lack of a computational cost analysis was criticized (k7fv); authors provided some computation time results in a rebuttal.
- The two data sets were considered non-standard and experiments on additional data such as a meta-dataset were desired (dA8b)
- The experiments on two data sets were considered overly limited, and experiments on larger scale datasets were desired (k7fv); author provided one additional data set (Mini-ImageNet) in a rebuttal
After discussion, reviewers felt the rebuttals addressed some of the concerns but some concerns such as how the method would work on datasets like meta-dataset remained.

Overall the paper feels borderline due to the amount of concerns that were raised, but reviewers mostly seemed to appreciate the rebuttals. Meta-learning is likely a rising need, as ever larger existing models become available whereas data gathered in individual scenarios/tasks may remain sparse; thus new probabilistic solutions to meta-learning and novel task detection could be valuable to the community. Although the approach is an adaptation of hierarchical Gaussian mixture modeling, authors have argued it contains various methodological novelties, and showing that it can yield competitive and outperforming results in meta-learning is useful. Therefore, it may be worth giving the work a chance; if the points raised in rebuttals are adequately incorporated into the paper, the work may be suitable for presentation at NeurIPS.